# Chemogenetic inactivation reveals the inhibitory control function of the prefronto-striatal pathway in the macaque brain

Mineki Oguchi [1,2], Shingo Tanaka[1,3], Xiaochuan Pan[4], Takefumi Kikusui [2], Keiko Moriya-Ito[5], Shigeki Kato[6], Kazuto Kobayashi [6] & Masamichi Sakagami [1✉]

The lateral prefrontal cortex (LPFC) has a strong monosynaptic connection with the caudate nucleus (CdN) of the striatum. Previous human MRI studies have suggested that this LPFC-CdN pathway plays an important role in inhibitory control and working memory. We aimed to validate the function of this pathway at a causal level by pathway-selective manipulation of neural activity in non-human primates. To this end, we trained macaque monkeys on a delayed oculomotor response task with reward asymmetry and expressed an inhibitory type of chemogenetic receptors selectively to LPFC neurons that project to the CdN. Ligand administration reduced the inhibitory control of impulsive behavior, as well as the task-related neuronal responses observed in the local field potentials from the LPFC and CdN. These results show that we successfully suppressed pathway-selective neural activity in the macaque brain, and the resulting behavioral changes suggest that the LPFC-CdN pathway is involved in inhibitory control.

[1] Brain Science Institute, Tamagawa University, Tokyo, Japan. [2] School of Veterinary Medicine, Azabu University, Kanagawa, Japan. [3] Department of Physiology, School of Medicine, Niigata University, Niigata, Japan. [4] Institute for Cognitive Neurodynamics, East China University of Science and Technology, Shanghai, China. [5] Tokyo Metropolitan Institute of Medical Science, Tokyo, Japan. [6] Department of Molecular Genetics, Institute of Biomedical Sciences, Fukushima Medical University, Fukushima, Japan. ✉email: sakagami@lab.tamagawa.ac.jp

Our brain makes value-based decisions through an inter-action between the prefrontal cortex (PFC) and the basal ganglia[1–4]. Anatomical studies using the macaque brain have revealed the existence of partially overlapping but distinct projections from each subarea of the PFC to the input nuclei of the basal ganglia, composing the striatum[5]. The lateral PFC (LPFC), which plays a central role in goal-directed behavior[6], is strongly associated with the head and body of the caudate nucleus (CdN) in the striatum[7–9]. While the anatomical connection from the LPFC to the CdN is unidirectional, the signal from the LPFC to the CdN is sent back to cortical areas via the thalamus, which constitutes part of the cortico-striato-thalamo-cortical loop[10]. Previous studies targeting the LPFC-CdN pathway have suggested different views on its function in decision-making, for instance, inhibitory control, especially in the sense of patience or self-control to prevent impulsive behavior[11–19], input and output gating of working memory[20–23], set shifting[24], model-based reinforcement learning[25], and category learning[2]. The vast majority of these findings have been mainly obtained by anatomical and functional connectivity analysis based on MRI including human participants. The PFC is a highly developed part of the brain in primates, and no homologous sites have been found in rodents, especially in regard to the LPFC[26]. In order to better understand the function of the LPFC-CdN pathway, it is necessary to study non-human primate models, especially macaque monkeys, using techniques that selectively regulate neural activity of a specific pathway. However, such physiological investigations to elucidate circuit function at a causal level have not been carried out until very recently due to the lack of appropriate technology applicable to the macaque brain.

To control neural activity of the LPFC in the macaque brain, a variety of techniques, such as aspiration and amputation[27], electric microstimulation[28], transcranial magnetic stimulation[29], local cooling[30], and administration of agonists and antagonists such as muscimol[31], have been traditionally used. These techniques are not pathway-selective and thus affect not only neurons that compose the target pathway but also those projecting through other off-target pathways. In recent years, perturbation of neural activity using optogenetics and chemogenetics has also been established in macaque monkeys (for review ref. [32]). Previous optogenetic studies have shown that pathway-selective regulation of neuronal activity is achieved by photo-stimulation of the axon terminals of opsin-expressing neurons[33,34]. The chemogenetic studies in macaque monkeys reported to date have not used pathway-selective control, as Designer Receptors Exclusively Activated by Designer Drugs (DREADDs) have been indiscriminately expressed in various neuronal types in a specific area, and exogenous ligands such as clozapine N-oxide (CNO) have been systemically delivered to the brain[35–40].

In rodents, several chemogenetic studies have reported pathway selectivity by directly delivering exogenous ligands to the brain by intracranial microinjection at axon terminals of DREADD-positive neurons[41,42]. An alternative approach widely used in rodent chemogenetics is to apply double virus transduction using the Cre/loxP system[43–45]. A series of macaque studies have used this method combined with the Tet-on system and tetanus neurotoxin to block synaptic transmission of selective spinal and subcortical pathways[46–49]. In our previous study, we tested whether this double virus transduction method is feasible to characterize the macaque prefrontal network (the pathway from the LPFC to the CdN and that from the LPFC to the frontal eye field) and observed doubly-transduced neurons in the LPFC[50]. However, that work was an anatomical study, and the ligand was not administered to the doubly-transduced monkeys.

In the present study, we applied chemogenetic double virus transduction to the macaque brain in order to selectively modulate LPFC neurons projecting to the CdN, using an inhibitory type of DREADDs (hM4D$_i$[51]). Two Japanese macaques were trained in an oculomotor delayed response task involving an asymmetric reward schedule. The behavioral and neural analyses suggest that the activity of neurons composing the LPFC-CdN pathway is suppressed by the chemogenetic manipulation, reducing the ability to inhibit impulsive behavior. This study achieves the successful regulation of neural activity using pathway-selective chemogenetic suppression in the macaque prefrontal network and reports changes both in behavior and neural activity, representing an important step forward in elucidating various neural circuit functions using non-human primates.

## Results

**Selective DREADDs expression to LPFC neurons projecting to the CdN.** To selectively suppress the activity of LPFC neurons projecting to the CdN, we used inhibitory DREADDs targeting the bilateral LPFC-CdN pathway in two macaque monkeys (Monkeys W and S). AAV5-hSyn-DIO-hM4D$_i$-mCherry was used as a local vector for the LPFC, and FuG-E(NeuRet)-nls/Cre-2A-eGFP was used as a retrograde vector for the CdN (Fig. 1a). AAV5 showed a high transduction efficiency in the primate brain among different AAV serotypes[52], while NeuRet selectively transduced neurons and was efficiently transported retrogradely[53–55]. To cover a wide range of target areas, we injected the virus solution to a total of 40 tracks of the bilateral LPFC of Monkey W and 48 tracks of that of Monkey S. We also injected a total of 40 and 46 tracks of the bilateral CdN of Monkey W and Monkey S, respectively (Fig. 1b). Injections were typically made at two different depths per track in the LPFC and at three different depths in the CdN. A 2.0 μL vector solution was injected at each depth (see "Methods"). The expression of mCherry, inserted as a marker of hM4D$_i$ expression, was observed in the LPFC of both monkeys (Fig. 1c, d and Supplementary Fig. 1a, b). One in four brain slices was immunohisto-chemically stained, and 3046 and 1241 mCherry-positive cells were identified in the left and right hemispheres of Monkey W, respectively, while the corresponding numbers in Monkey S were 4420 and 3521 cells. The mCherry-positive cells were found mainly within the bank of the principal sulcus (PS) and its dorsal and ventral regions. High proportion of eGFP-positive cells (Monkey S; Fig. 1e, f) and needle traces (Monkey W; Supplementary Fig. 1c, d) were observed in the CdN. These results suggest that hM4D$_i$ was selectively expressed in LPFC neurons that project to the CdN.

**CNO administration during the one-direction reward saccade task.** Before vector injection, the monkeys were trained on the one-direction reward saccade task (1DR task[56,57], which is a memory-guided saccade task involving an asymmetric reward schedule (Fig. 2a). A trial started when the monkeys fixated on the central fixation point (FP). A cue stimulus was then presented pseudo-randomly to either the left or the right for a short period of time. After about a 1 s of delay, the FP disappeared, and the monkeys used this as the go signal to make a saccade to the previously cued position. If the monkeys made the correct saccade, a drop of water was given as a reward accompanied with a high-frequency tone. The amount of the reward for the left and right was different and switched pseudo-randomly block by block. For instance, in one block, the left direction was associated with a large reward, whereas the right direction was associated with a small reward, and in the next block, vice versa. The correction method was used for this task; the same trial was repeated if the monkeys made an error. Thus, monkeys had to succeed even on small-reward trials with low motivation in order to

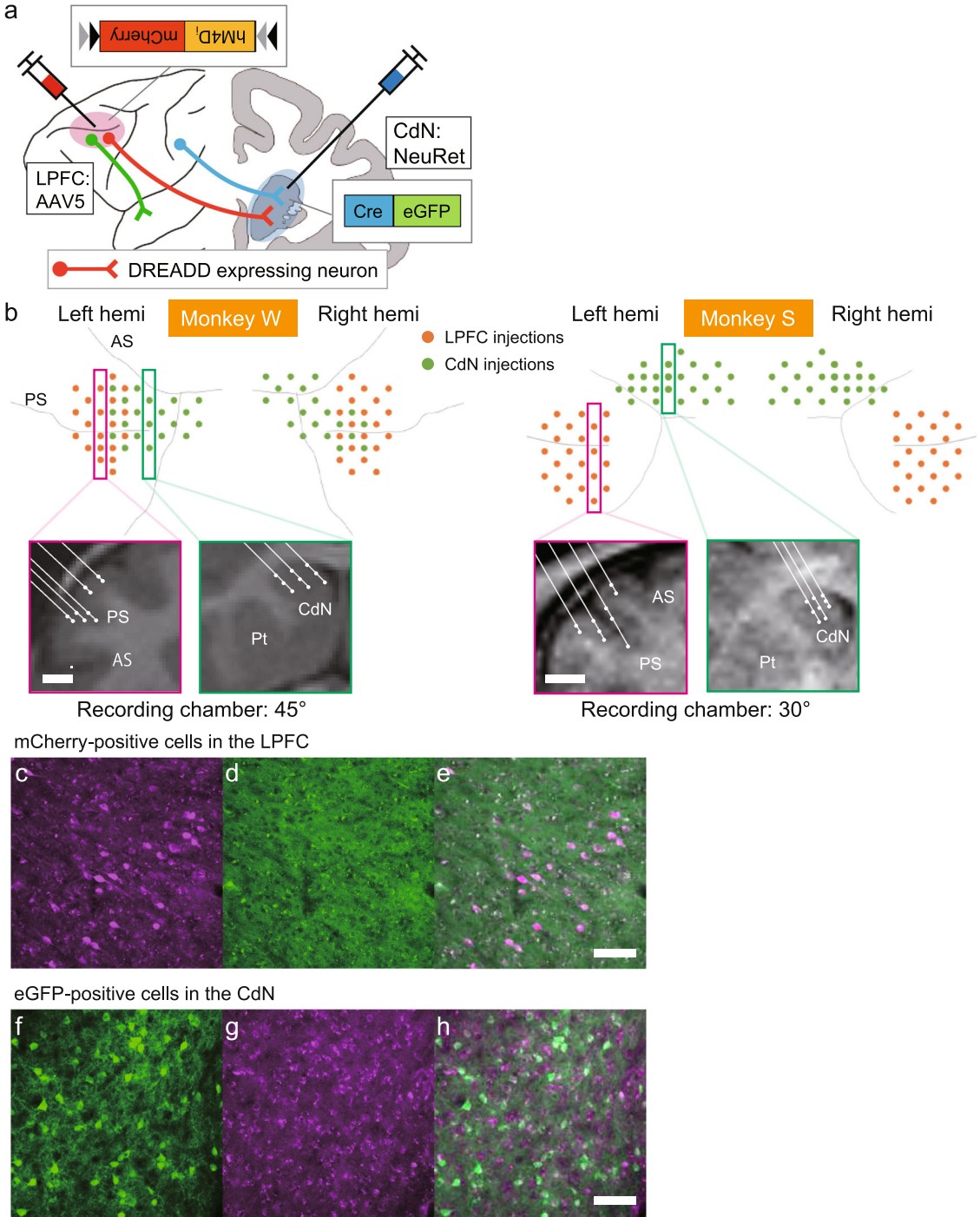

**Fig. 1 Chemogenetic double virus transduction and injection sites in the LPFC and CdN. a** Illustration of the chemogenetic double virus transduction method. The locally transducing virus (AAV5-hSyn-DIO-hM4D$_i$-mCherry) was injected into the LPFC, while the retrograde virus (FuG-E(NeuRet)-nls/Cre-2A-eGFP) was injected into the CdN. Only doubly-transduced neurons whose cell bodies were in the LPFC and axon terminals in the CdN expressed the *hM4D$_i$-mCherry* gene. The activity of these doubly-transduced neurons could be suppressed by CNO administration. **b** Injection sites of the two viruses. Above: View of virus injection tracks from the diagonal direction for Monkey W (left) and Monkey S (right). Orange dots indicate AAV injection tracks in the LPFC. Green dots indicate NeuRet injection tracks in the CdN. Below: Representative 2D coronal reconstructions of injection positions (white dots) in the LPFC and CdN, corresponding to the tracks boxed in the upper panels. Scale bar: 5 mm. **c** Doubly-transduced, mCherry-positive neurons (magenta) in the right LPFC as observed using a WIG filter cube and **d** the micrograph of the same area as (**c**) as observed using a NIBA filter cube. **e** The superimposed image of (**c**) and (**d**). Scale bar: 100 μm. **f** eGFP-positive neurons (green) in the ipsilateral CdN of Monkey S. **g** Micrograph of the same area as in (**f**). **h** The superimposed image of (**f**) and (**g**). Scale bar: 100 μm. AAV adeno-associated virus, LPFC lateral prefrontal cortex, PS principal sulcus, AS arcuate sulcus, CdN caudate nucleus, Pt putamen, DREADD Designer Receptors Exclusively Activated by Designer Drugs.

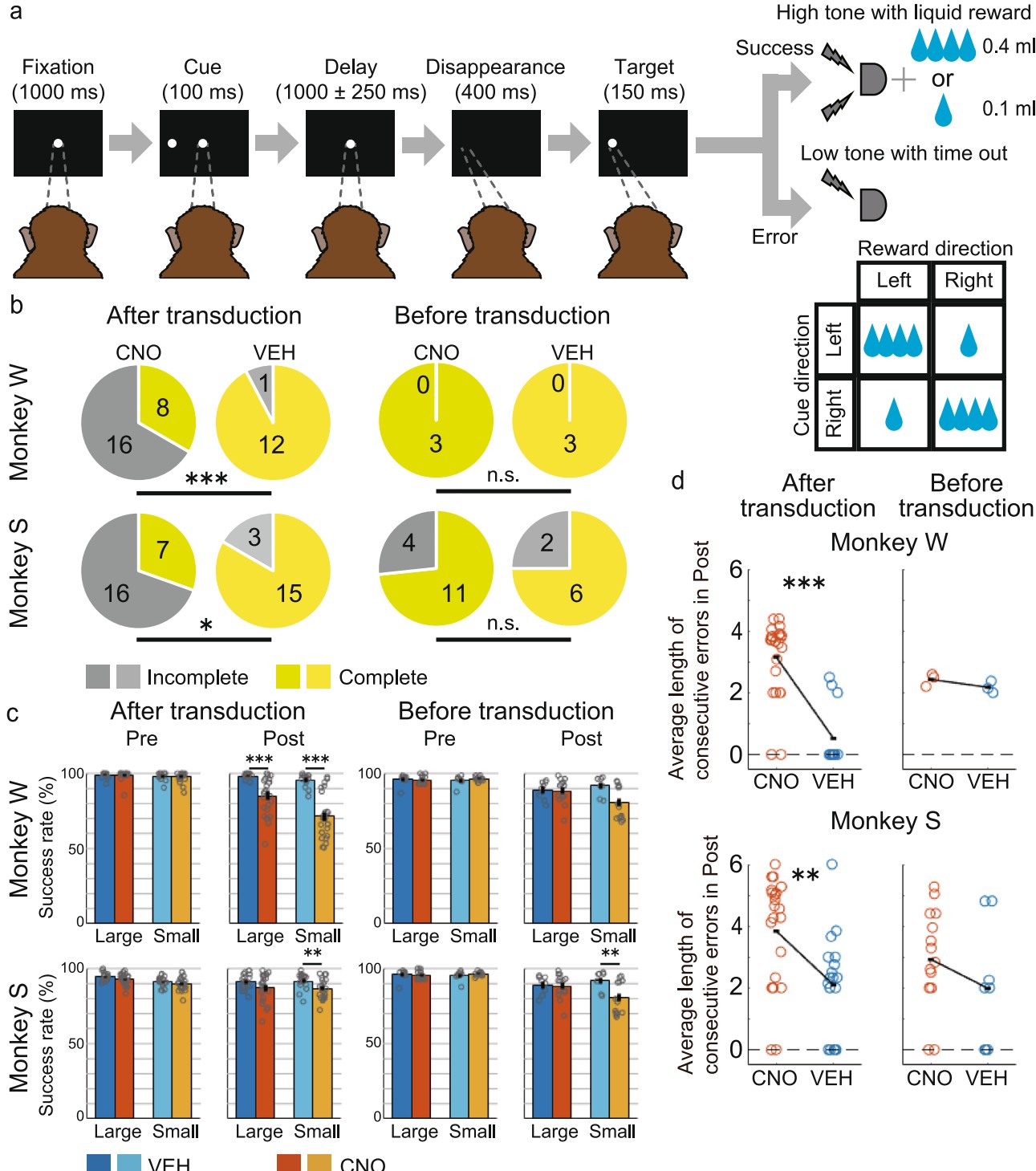

**Fig. 2 Oculomotor delayed response task and behavioral effects of CNO administration on session completion and error frequency. a** Time sequence of the one-direction reward (1DR) saccade task. A memory cue was briefly presented in either the left or the right direction. Within a block of trials, a large reward (0.4 ml of water) was consistently associated with one direction, whereas a small reward (0.1 ml) was associated with the opposite direction. The association between reward size and direction was pseudo-randomly changed in each block without any cue. **b** Proportion of complete and incomplete sessions before (right) and after (left) double virus transduction. Upper: Monkey W. Lower: Monkey S. Yellowish colors indicate complete sessions, and greyish colors indicate incomplete sessions. *$P < 0.05$, **$P < 0.01$, ***$P < 0.001$, n.s. non-significant (common to all figures). **c** Success rate on the first 160 trials before CNO/VEH administration (Pre) and on the last 160 trials after CNO/VEH administration (Post), separately calculated for large- and small-reward trials for Monkey W (upper) and Monkey S (lower). Each dot indicates the success rate per session for each condition. VEH condition: bluish bars; CNO condition: Yellowish bars. Error bars: standard error of the mean. **d** Average length of consecutive errors during Post before (right panels) and after (left panels) double virus transduction. Each dot indicates the average length of consecutive errors per session. CNO clozapine N-oxide, VEH vehicle.

obtain future large rewards. One block consisted of 40 non-correction trials.

In a typical session, a venous line was established from the leg at the beginning, and the monkeys first performed 160 trials of the 1DR task. Subsequently, the DREADD ligand CNO (5 mg/kg BW) or a control solution without CNO (vehicle: VEH) was administered systemically intravenously. The monkeys were then required to perform 720 trials. During the session, neural activity was recorded using two linear multi-contact electrodes with 16 channels simultaneously from the LPFC and the ipsilateral CdN.

**CNO administration increases the rate of incomplete sessions and consecutive errors**. We first analyzed the effect of CNO administration on behavior to test the suppressive effect on CdN-projecting LPFC neurons. Previous studies showed that, in macaque monkeys, the concentration of CNO in the cerebrospinal fluid gradually increases after systemic administration, reaching its peak after 60 min and persisting for several hours[37,58]. A complete session was defined when monkeys successfully completed 720 trials (about 2.5 h) after CNO/VEH administration. If the monkeys made consecutive errors that exceeded the threshold after CNO/VEH administration, we stopped the session before reaching 720 trials. This was defined as an incomplete session (see "Methods"). In the sessions after the double virus transduction, the proportion of incomplete sessions was significantly greater for both monkeys in the CNO than in the VEH condition (Fig. 2b; left, % incomplete, monkey W: CNO 66.7% vs. VEH 7.7 %, $p < 0.001$; Monkey S: CNO 69.6% vs. VEH 16.7%, $p = 0.022$, Fisher's exact test). Incomplete sessions in the CNO condition typically resulted in a gradual increase of consecutive errors after more than 1 h from CNO administration, and the monkeys could no longer return to their normal performance (Supplementary Fig. 2). To check the behavioral effect of CNO itself, we performed behavioral tests after CNO/VEH administration but prior to the double transduction. In the pre-transduction behavioral tests, although the delay period after the cue onset was set to about 3 s, which was longer than about 1 s in the post-transduction sessions (see "Methods"), the proportion of incomplete sessions was lower and not significantly different between the CNO and VEH conditions (Fig. 2b; right, % incomplete, Monkey W: CNO 0% vs VEH 0%, $p = 1.00$; Monkey S: CNO 26.7% vs. VEH 25.0%, $p = 1.00$, Fisher's exact test). An increase in the proportion of incomplete sessions was thus specific to the CNO condition after the double transduction.

We next calculated the success rates in the first 160 trials before CNO/VEH administration (Pre) and the last 160 trials after CNO/VEH administration (Post). Here, to account for the effect on consecutive errors, repetitive error trials and correction success trials were also included in the calculation. In general, the success rate gradually declined toward the end of the session due to satiation and fatigue. We hence compared the correct rates in the CNO and VEH conditions separately for Pre and Post. After the double transduction, in Pre, there was no significant difference in the success rates between the CNO and VEH condition for both the large- and small-reward trials in Monkey W (Fig. 2c; large: $t(35) = 0.18$, $p = 0.862$; small: $t(35) = 0.36$, $p = 0.719$. two-sample $t$-test) and Monkey S (large: $t(40) = 1.62$, $p = 0.112$; small: $t(40) = 0.97$, $p = 0.337$). In contrast, in Post, the success rates were significantly lower in the CNO than in the VEH condition for both the large- and small-reward trials in Monkey W (large: $t(35) = 3.76$, $p < 0.001$; small: $t(35) = 5.74$, $p < 0.001$). The success rate was significantly lower in the CNO than in the VEH condition only for small-reward trials in Monkey S (large: $t(40) = 1.77$, $p = 0.085$; small: $t(40) = 2.70$, $p = 0.010$). The interaction between the drug (CNO, VEH) and

time (Pre, Post) was not significant either for Monkey W ($F(1,144) = 0.95$, $p = 0.334$, two-way analysis of variance (ANOVA)) or for Monkey S ($F(1,164) = 0$, $p = 0.997$). In the behavioral test prior to the double transduction, there was no significant difference in any of the conditions in Monkey W (Pre, Large: $t(4) = 1.50$, $p = 0.208$; small: $t(4) = 0.93$, $p = 0.404$; Post, large: $t(4) = 0.82$, $p = 0.458$; small: $t(4) = 0.67$, $p = 0.542$). In contrast, in Monkey S, the success rate was lower in the CNO than in the VEH condition in small-reward trials in Post, as in the sessions after the double transduction (Pre, large: $t(21) = 0.49$, $p = 0.628$; small: $t(21) = 1.05$, $p = 0.304$; Post, large: $t(21) = 0.22$, $p = 0.831$; Small: $t(21) = 2.69$, $p = 0.014$). The interaction between the drug and time was not significant either for Monkey W ($F(1,20) = 0.10$, $p = 0.755$) or for Monkey S ($F(1,88) = 1.40$, $p = 0.240$). The success rate was thus reduced after CNO administration compared to that after VEH administration specifically after the double transduction in Monkey W. In Monkey S, the success rate decreased in small-reward trials after the transduction, but this was also observed even before the transduction.

To assess the effect of CNO administration on consecutive errors, we calculated the average length of consecutive errors per session during the Post period. We found that the value increased in the CNO condition compared to that in the VEH condition in both monkeys (Fig. 2d; mean, Monkey W: CNO 3.15 vs. VEH 0.52, $p < 0.001$; Monkey S: CNO 3.83 vs. VEH 2.17, $p = 0.006$, Wilcoxon rank-sum test). In contrast, there was no significant difference in the average length between the CNO and VEH conditions prior to vector injection in both monkeys (Monkey W: CNO 2.43 vs. VEH 2.17, $p = 0.200$; Monkey S: CNO 2.92 vs. VEH 1.98, $p = 0.228$). These results showed that suppression of CdN-projecting LPFC neurons increases the rate of incomplete sessions and the frequency of consecutive errors.

In order to examine the period during which the frequency of errors changed, we divided errors during the Post condition into four types depending on the time when they occurred: refusal to initiate a fixation before the fixation start, fixation break before the cue onset, fixation break during the delay period, and saccade error after the go signal. We then calculated the frequency of errors in each type separately (see "Methods"). In Monkey W, the frequency of errors increased for all types of errors in CNO condition, and only for fixation breaks during the delay period with a smaller difference in VEH condition, when Pre- and Post-trials were compared (Supplementary Fig. 3). In Monkey S, the frequency of saccade errors decreased in both the CNO and VEH conditions, but all other error types increased only in the CNO condition. Although there was an inconsistency between monkeys with respect to the frequency of saccade errors, errors related to fixation breaks seemed to increase after CNO administration in both monkeys.

**CNO administration causes impulsive oculomotor behavior**. In the 1DR task, monkeys were required to make a saccade to the pre-cued position immediately after the FP offset. To investigate the effect of the suppression of CdN-projecting LPFC neurons on this saccade eye movement, we analyzed several aspects of saccade behavior. Previous studies have suggested that the LPFC-CdN pathway is involved in input and output gating to working memory[20–23]. If CNO administration caused a working memory dysfunction, we would expect that monkeys would be unable to reflect the information of the cue position in their behavior and, therefore, to increase erroneous saccades to the opposite target window. The results were contrary to this prediction. Among trials in which a saccade was directed to the left or right target window, the percentage of those with saccades in the opposite

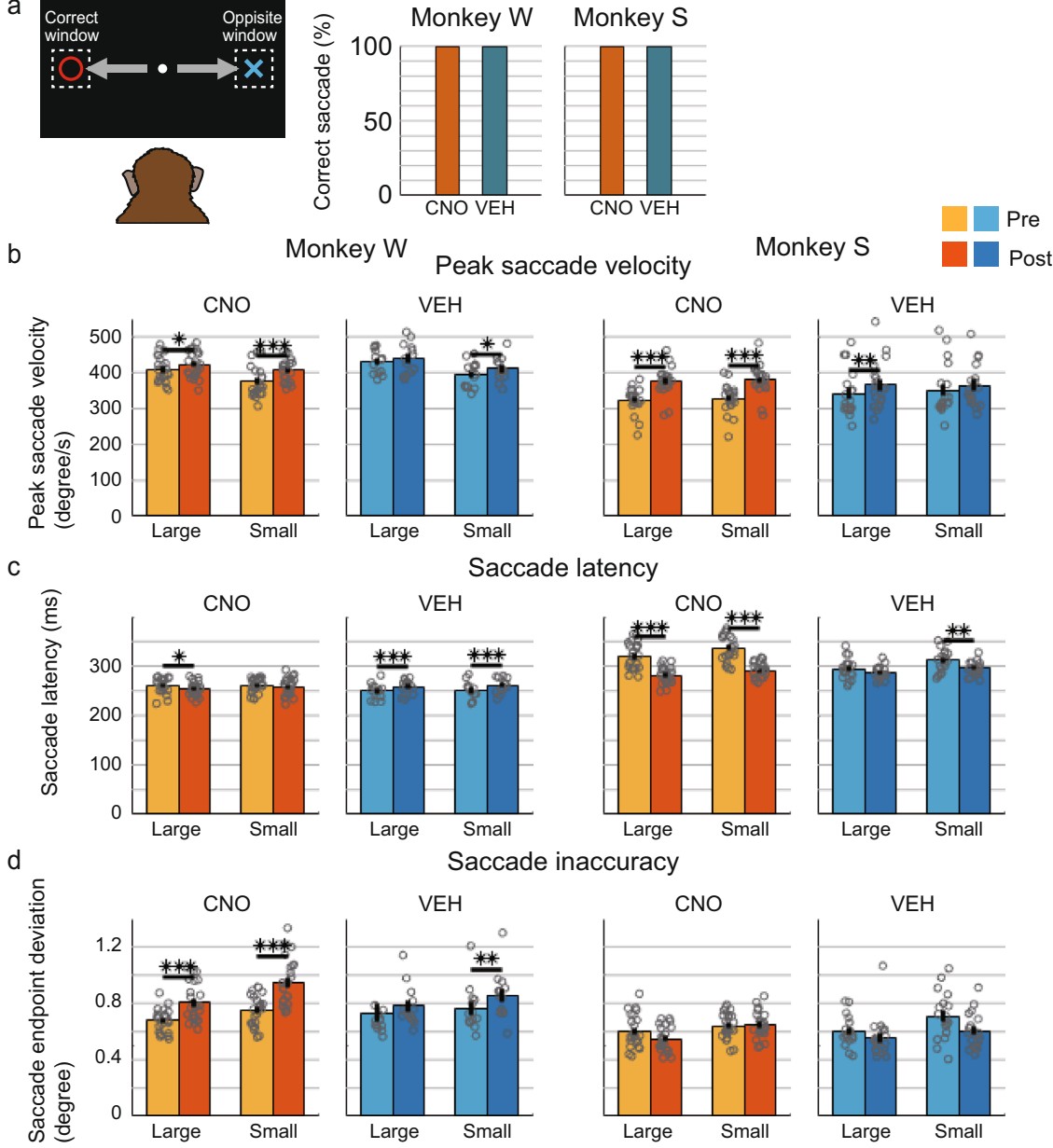

**Fig. 3 Change in saccade behavior after the go signal. a** Percentage of correct saccade trials among trials in which the monkeys made a saccade either into the correct target window or into the opposite window in the Post period for Monkey W (left) and Monkey S (right). **b** Peak saccade velocity after the go signal (i.e., after FP offset). Pale colors indicates peak saccade velocity on Pre and deep colors indicates that on Post, separately calculated for large and small reward trials for Monkey W (left) and Monkey S (right). Each dot indicates the mean value per session for each condition. Error bar: standard error of the mean. **c** Saccade latency (the time from FP offset to the saccade start). **d** Saccade inaccuracy (the distance between saccade endpoint and the center of the target). FP: fixation point.

direction was very small, and there was no significant difference between the CNO and VEH conditions for both monkeys (Fig. 3a; % correct saccades in the Post period, Monkey W: CNO 99.73% vs. VEH 99.90%, $p = 0.212$; Monkey S: CNO 99.81% vs. VEH 99.96%, $p = 0.136$, Fisher's exact test).

We next analyzed the effect of CNO administration on the speed, latency, and accuracy of saccadic eye movements. Here, we used only eye movement data from non-correction correct trials. Peak saccade velocity was significantly higher on Post than on Pre in both large- and small-reward trials after CNO administration in both Monkey W (Fig. 3b; large: $t(22) = 2.74$, $p = 0.012$; small: $t(22) = 4.46$, $p < 0.001$, paired samples $t$-test) and Monkey S (large: $t(22) = 10.84$, $p < 0.001$; small: $t(22) = 11.60$, $p < 0.001$).

After VEH administration, peak saccade velocity was significantly higher only in small-reward trials in Monkey W (large: $t(12) = 1.93$, $p = 0.077$; small: $t(12) = 2.62$, $p = 0.023$) and large-reward trials in Monkey S (large: $t(17) = 3.29$, $p = 0.004$; small: $t(17) = 1.18$, $p = 0.254$) with smaller differences. The interaction between the drug and time was not significant for Monkey W ($F(1,140) = 0.60$, $p = 0.439$, two-way ANOVA) but significant for Monkey S ($F(1,160) = 3.95$, $p = 0.049$).

Saccade latency was significantly shorter in the Post than in the Pre period in large-reward trials after CNO administration (Fig. 3c; large: $t(22) = 2.40$, $p = 0.025$; small: $t(22) = 0.95$, $p = 0.354$) but significantly longer in both large- and small-reward trials after VEH administration (large: $t(12) = 5.26$,

$p < 0.001$; small: $t(12) = 4.77$, $p < 0.001$; paired samples $t$-test) in Monkey W. In Monkey S, saccade latency was significantly shorter in the Post than in the Pre period in both large- and small-reward trials in the CNO condition (large: $t(22) = 11.01$, $p < 0.001$; small: $t(22) = 12.09$, $p < 0.001$) and only in small-reward trials in the VEH condition (large: $t(17) = 1.87$, $p = 0.079$; small: $t(17) = 3.13$, $p = 0.006$). The interaction between the drug and time was significant both for Monkey W ($F(1,140) = 7.21$, $p = 0.008$, two-way ANOVA) and for Monkey S ($F(1,160) = 19.4$, $p < 0.001$). The above results showed that peak saccade velocity became relatively faster and saccade latency became relatively shorter after CNO administration in post-transduction sessions, suggesting that suppression of CdN-projecting LPFC neurons result in hasty or impulsive saccades after the go signal. This change in saccade impulsivity progressed gradually after CNO administration during both complete and incomplete sessions, followed by a sharp increase in saccade latency just near the end of the sessions, especially in incomplete sessions (Supplementary Fig. 4). An increase in errors was observed shortly before this rise in saccade latency during the incomplete sessions.

Next, we analyzed the changes in peak saccade velocity and saccade latency during the test sessions before double virus transduction as an additional control. Note, however, that it is difficult to exactly compare the changes before and after transduction because the delay before the transduction is approximately 3 s, which is longer than that in after transduction (approximately 1 s). For Monkey W, the peak saccade velocity was similarly larger on Post than on Pre in both large- and small-reward trials and in both CNO (Supplementary Fig. 5a; large: $t(4) = 8.71$, $p = 0.013$; small: $t(4) = 6.37$, $p = 0.037$, paired samples $t$-test) and VEH (large: $t(4) = 5.05$, $p = 0.024$; small: $t(4) = 4.89$, $p = 0.039$) conditions. For Monkey S, there was no significant difference between Pre and Post for the CNO (large: $t(28) = 1.97$, $p = 0.059$; small: $t(28) = 1.08$, $p = 0.287$) and VEH (large: $t(14) = 1.81$, $p = 0.091$; small: $t(14) = 0.18$, $p = 0.861$) conditions. The interaction between the drug and time was not significant for Monkey W ($F(1,20) = 0$, $p = 0.970$, two-way ANOVA) or Monkey S ($F(1,88) = 0.08$, $p = 0.776$). Therefore, there was no clear difference in the peak saccade velocity for the CNO and VEH conditions before transduction unlike after transduction. During the pre-transduction sessions, there was no significant difference in the saccade latency between Pre and Post under the CNO condition for Monkey W (Supplementary Fig. 5b; large: $t(4) = 0.05$, $p = 0.965$; small: $t(4) = 0.32$, $p = 0.768$), and the saccade latency was shorter in Post than in Pre under VEH condition (large: $t(4) = 6.20$, $p = 0.003$; small: $t(4) = 8.38$, $p = 0.001$), which was opposite the trend after transduction. For Monkey S, there were no significant differences associated with the CNO (large: $t(28) = 0.11$, $p = 0.911$; small: $t(28) = 0.64$, $p = 0.530$) or VEH (large: $t(14) = 1.11$, $p = 0.254$; small: $t(14) = 1.20$, $p = 0.250$) conditions. The interaction between the drug and time was significant for Monkey W ($F(1,20) = 14.4$, $p = 0.001$, two-way ANOVA) but not for Monkey S ($F(1,60) = 0.48$, $p = 0.490$). Therefore, there was no similar trend in the change in saccade latency under the CNO conditions before and after the double virus transduction.

**Effects of CNO administration on saccade accuracy**. Next, to analyze the accuracy of saccades, we calculated the deviation of the endpoints of the saccade from the center of the target. The deviation was greater in the Post than in the Pre period in both large- and small-reward trials after CNO administration (Fig. 3d; large: $t(22) = 5.06$, $p < 0.001$; small: $t(22) = 5.44$, $p < 0.001$, paired samples $t$-test), and only in small-reward trials after VEH administration (large: $t(12) = 1.05$, $p = 0.313$; small: $t(12) = 3.55$,

$p = 0.004$) with smaller differences in Monkey W. The deviation was not significantly different between Pre and Post in any conditions in Monkey S (CNO, large: $t(22) = 1.83$, $p = 0.080$; small: $t(22) = 0.18$, $p = 0.861$; VEH, large: $t(17) = 0.98$, $p = 0.340$; small: $t(17) = 1.99$, $p = 0.063$). The interaction between the drug and time was not significant either for Monkey W ($F(1,140) = 3.17$, $p = 0.077$, two-way ANOVA) or for Monkey S ($F(1,160) = 1.49$, $p = 0.224$). Therefore, the saccade accuracy was worse after CNO administration in Monkey W. This result was consistent with the fact that the frequency of saccade errors increased after CNO administration only in Monkey W (Supplementary Fig. 3). However, there was a tendency for saccade accuracy to decrease for the CNO compared with VEH, even for the pre-transduction analysis. For Monkey W, the accuracy deteriorated significantly in small reward trials for both CNO (Supplementary Fig. 5c; large: $t(4) = 8.71$, $p = 0.281$; small: $t(4) = 6.37$, $p < 0.001$, paired samples t-test) and VEH (large: $t(4) = 5.05$, $p = 0.050$; small: $t(4) = 4.89$, $p = 0.013$) conditions, with a larger difference in CNO condition. For Monkey S, accuracy deteriorated in both large and small reward trials for the CNO condition (large: $t(28) = 2.78$, $p = 0.010$; small: $t(28) = 4.27$, $p < 0.001$) and only in small reward trials for the VEH condition (large: $t(14) = 0.62$, $p = 0.542$; small: $t(14) = 4.81$, $p < 0.001$). The interaction between the drug and time was not significant for Monkey W ($F(1,20) = 0.01$, $p = 0.934$) or Monkey S ($F(1,88) = 0.12$, $p = 0.725$). Therefore, the deterioration of saccade accuracy may be due to CNO administration per se.

**Effects of CNO administration on eye movement behavior at different time periods**. As shown above, the saccades before reward delivery became more impulsive after CNO administration. This could be interpreted as an increase in reward impulsivity, elicited as impatient behavior toward the reward. We next analyzed the rate of fixation errors induced by the cue stimuli to investigate the effect of CNO administration on a different type of impulsivity, that is, stimulus impulsivity, elicited as reactive behavior toward salient stimuli. Among fixation break errors during the delay period, the proportion of those produced by making a saccade to the cue direction after the cue onset was not significantly different between the CNO and VEH conditions for both Pre and Post periods and for both Monkey W (Supplementary Fig. 6a; % cue-induced error, Pre: CNO 16.3% vs. VEH 20.8%, $p = 0.743$; Post: CNO 27.4% vs. VEH 31.9%, $p = 0.496$, Fisher's exact test) and Monkey S (Pre: CNO 1.11% vs. VEH 1.49%, $p = 1.000$; Post: CNO 0.63% vs. VEH 0%, $p = 1.000$), indicating that CNO administration does not increase stimulus impulsivity.

For further comparison, we analyzed saccadic eye movements at different time points when the monkeys made a saccade to the FP at the start of the trial. It was difficult to accurately estimate saccade latency after FP onset, because the time limit after FP onset was considerably longer than after the go signal (5000 ms vs. 550 ms), and the eye position and speed at the timing of FP onset were considerably diverse across trials. Instead, we calculated the peak saccade velocity as the eyes headed toward the FP using trials where the eye position had been away from the FP window at FP onset. Peak saccade velocity was not significantly increased but was rather decreased after CNO administration in Monkey W (Supplementary Fig. 6b; CNO: $t(22) = 2.90$, $p = 0.008$; VEH: $t(12) = 0.34$, $p = 0.740$, paired samples $t$-test) and was unchanged in Monkey S (CNO: $t(22) = 0.13$, $p = 0.895$; VEH: $t(17) = 1.00$, $p = 0.327$). As shown above, peak saccade velocities increased in both monkeys under the CNO condition after the go signal, but this was not observed in saccades at the trial start. These comparisons indicated that the

increase in impulsivity after CNO administration is specific to saccades after the go signal, rather than a general effect.

**CNO administration decreases positive cue response in LFP power in the LPFC.** We recorded LFPs simultaneously from the ipsilateral LPFC and CdN using two 16-channel multi-contact electrodes (U-probe) during the experimental sessions. In total, we obtained 752 channels of LFPs from the LPFC (368 channels from Monkey W and 384 channels from Monkey S) and the same number of channels from the CdN. We selected 676 LPFC channels (89.9%) and 660 CdN channels (87.8%) for subsequent analyses by removing channels with excessively low signal levels or high noise levels (see "Methods").

To investigate the effect on task-related signals from the LPFC, we first aligned the LFP signal to the cue onset and performed a time-frequency analysis to make spectrograms. Only channels that showed significant cue response were used in this analysis (655 out of 676 channels; see "Methods"). Particularly strong positive cue responses were observed in frequencies centered in the beta band in Monkey W and in wider frequencies centered in the gamma band in Monkey S (Fig. 4a, see especially the Pre-large panels). To check if the strength of the cue response was modulated by the predicted reward size in Pre, we extracted time-frequency regions of interest (ROIs) that showed strong cue responses based on a cluster-based non-parametric test[59,60] (see "Methods"), separately for large- and small-reward trials, and averaged the values of the data points that composed the ROIs. In both monkeys, cue responses in Pre were significantly stronger in large- than in small-reward trials (Monkey W: $t(335) = 8.63$, $p < 0.001$; Monkey S: $t(322) = 18.79$, $p < 0.001$, paired samples $t$-test).

To examine the effect of CNO administration on these reward-modulated cue responses, we next extracted ROIs by comparing large- and small-reward trials in Pre (combined CNO and VEH conditions). We calculated the difference between Pre and Post in the power of the ROIs and statistically analyzed the results between the CNO and VEH conditions. There were significant differences in large-reward trials in Monkey W (Fig. 4b, left; large: $t(334) = 5.08$, $p < 0.001$; small: $t(334) = 1.01$, $p = 0.312$, two-sample $t$-test) and in both large- and small-reward trials in Monkey S (Fig. 4b, right; large: $t(321) = 8.30$, $p < 0.001$; small: $t(321) = 3.26$, $p = 0.001$), with decreases in the power of the ROIs after CNO administration.

**CNO administration decreases positive cue response in LFP power in the CdN.** We next performed the same analysis on the LFP signal recorded from the CdN. Only cue-responsive channels were used in this analysis (641 out of 660 channels). Positive cue responses were found in the same frequency band as in the LPFC for each monkey (Fig. 5a). Cue responses were significantly stronger in large- than in small-reward trials (Monkey W: $t(334) = 6.41$, $p < 0.001$; Monkey S: $t(305) = 25.15$, $p < 0.001$, paired samples $t$-test). ROI analysis showed significant differences both in large- and small-reward trials in Monkey W (Fig. 5b, left; large: $t(333) = 3.76$, $p < 0.001$; small: $t(333) = 3.82$, $p < 0.001$, two-sample $t$-test) and in large-reward trials in Monkey S (Fig. 5b, right; large: $t(304) = 8.07$, $p < 0.001$; small: $t(304) = 0.79$, $p = 0.429$), with decreases in the power of the ROIs after CNO administration. These results suggest that suppression of CdN-projecting LPFC neurons reduces the LFP power in the cue response in both the LPFC and CdN.

Previous studies have shown that a large proportion of LPFC neurons respond more strongly to visual stimuli presented in the contralateral visual field (e.g., ref. [61]). To examine lateral differences in the CNO effect, we divided the recorded LFP data

into trials in which the visual stimuli were presented on the contralateral side and those in which they were presented on the ipsilateral side of the recording site, and performed the same time-frequency analysis as above separately for contralateral and ipsilateral conditions. We observed similar results in both contralateral and ipsilateral trials as before the division. In both cases, the cue response was particularly strong in large-reward trials and weakened only in the CNO condition (Supplemental Figs. 7–10). In addition, we analyzed the LFP data aligned to the fixation point onset. There was no clear response for any condition in either monkey as there was after the cue onset (Supplementary Fig. 11).

**Consistency of CNO effect on cue response between the two monkeys.** In the above time-frequency analysis, the attenuation of the cue response after CNO administration was detected in the beta band in Monkey W and in the gamma band in Monkey S, indicating inconsistency between the two monkeys. The cue response in Monkey S, however, spanned a wide frequency range (Figs. 4a and 5a), and it is possible that the effect of CNO administration was also detected in the beta band. Therefore, we applied the ROI around the beta band identified in Monkey W to Monkey S and, conversely, the ROI around the gamma band identified in Monkey S to Monkey W, with the time adjusted to the onset of the original ROI, and performed the same analysis as above to examine the effects of CNO administration on different frequency bands. Around the gamma band of the spectrograms obtained from Monkey W, the effect of CNO administration was not observed in the LPFC (Fig. 6a, left: large: $t(334) = 0.093$, $p = 0.926$; small: $t(334) = 0.319$, $p = 0.750$). The effect of CNO administration was smaller than that of VEH administration in large-reward trials and larger in small-reward trials in the CdN (Fig. 6b, left; large: $t(333) = 3.23$, $p = 0.001$; small: $t(333) = 2.27$, $p = 0.024$). Around the beta band of the spectrograms obtained from Monkey S, the effect of CNO administration was significantly larger than that of VEH administration in both large- and small-reward trials in the LPFC (Fig. 6a, right; large: $t(321) = 9.38$, $p < 0.001$; small: $t(321) = 3.97$, $p < 0.001$) and in large-reward trials in the CdN (Fig. 6b, right; large: $t(304) = 4.45$, $p < 0.001$; small: $t(304) = 0.55$, $p = 0.585$). These results indicate that the effect of CNO administration was consistent between the two monkeys in the beta band.

**CNO administration reduces negative saccade response in LFP power in the LPFC.** Next, to test the effect of CNO administration on neural activity during saccades, we performed a spectrogram analysis of the LFP signal from the LPFC, which was aligned to the go signal (FP offset). There was no positive response as observed after the cue onset, but there was a characteristic negative component above the beta band (Fig. 7a).

In order to quantitatively analyze the effect of CNO on this negative component, we extracted ROIs using cluster analysis with the data from 0 to 500 ms before the cue onset as a baseline. Here, we split the data into large- and small-reward trials and calculated the ROI for each. The differences in the ROI signal before and after CNO administration (Pre–Post) were compared with the corresponding differences after VEH administration. In Monkey W, the negative response was attenuated after CNO administration and was significantly different from that after VEH administration in both large- and small-reward trials (Fig. 7b, left; large: $t(334) = 3.79$, $p < 0.001$; small: $t(334) = 4.08$, $p < 0.001$, two-sample $t$-test). In Monkey S, the negative response was significantly attenuated in large-reward trials (Fig. 7b, right; large: $t(321) = 2.64$, $p = 0.009$; small: $t(321) = 1.44$, $p = 0.151$, two-sample $t$-test).

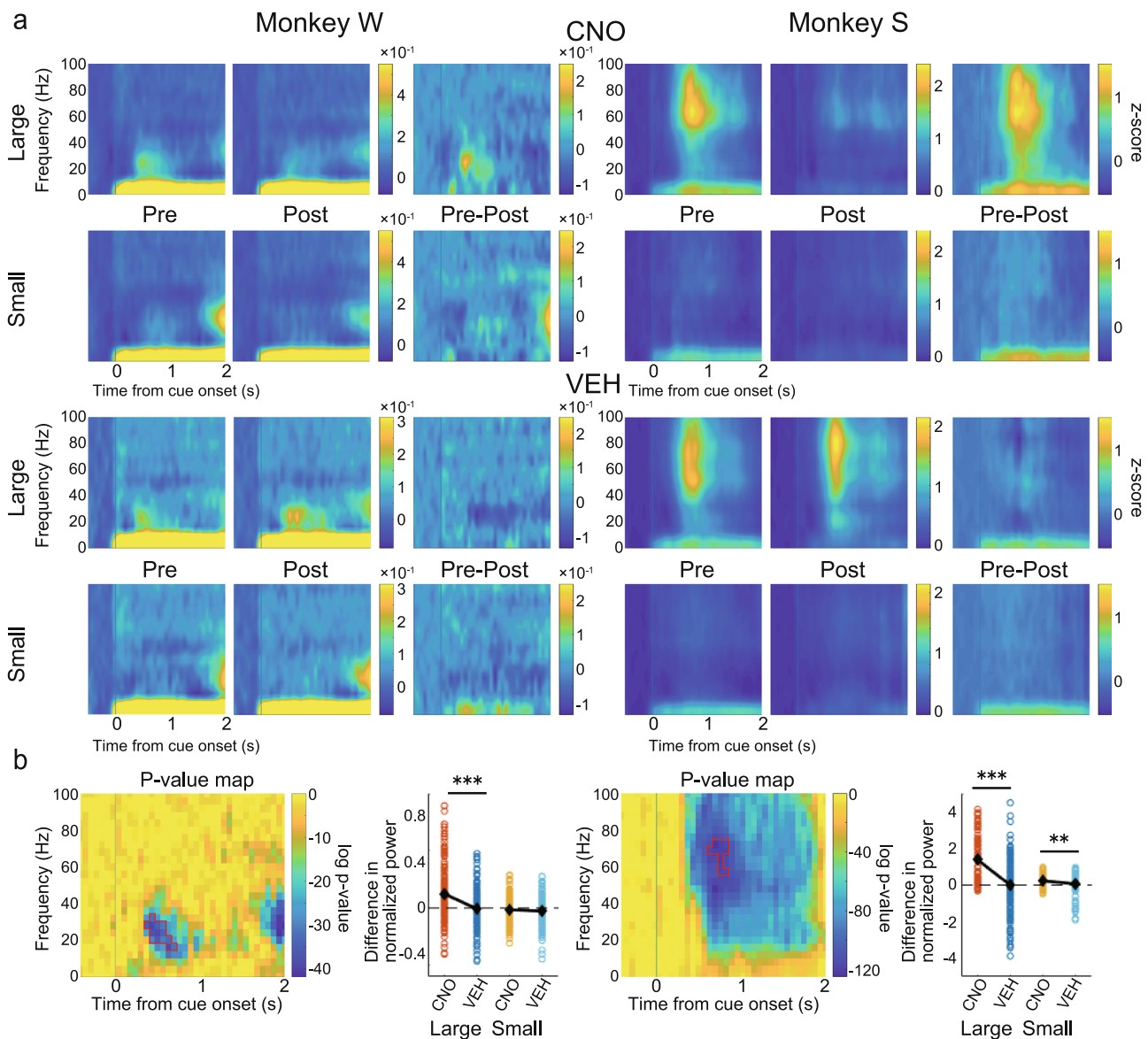

**Fig. 4 Effects on spectrogram power in the LPFC after cue onset. a** Normalized and averaged spectrograms of LFPs from electrodes in the LPFC for Monkey W (Left) and Monkey S (Right), divided into large- and small-reward trials in the CNO (upper) and VEH (lower) condition. From left to right for each monkey, each spectrogram referred to Pre, Post, and their subtraction (Pre–Post). Color represents normalized power (−500 to 0 ms as baseline). Time 0 refers to the cue onset. **b** For each monkey: left: *p*-value map generated to extract time-frequency ROIs for statistical testing. Color represents log *p*-value obtained by comparing spectrograms in large- and small-reward trials. Areas surrounded by red lines indicate ROIs, which are the largest cluster among time-frequency data points having *p*-value in the bottom 5% (see "Methods"); right: comparison of the effects of CNO/VEH administration (Pre–Post) on normalized power averaged within ROIs divided into large- and small-reward trials. Each dot refers to the subtracted power from each electrode channel. Black diamonds represent mean values. CNO clozapine N-oxide, VEH vehicle, LFP local field potential, LPFC lateral prefrontal cortex, ROI region of interest.

**CNO administration reduces negative saccade response in LFP power in the CdN.** We performed the same analysis for the LFP signal from the CdN. The results also showed a negative component after the go signal (Fig. 8a). Statistical analysis showed a significant attenuation of the negative response after CNO administration in small-reward trials in Monkey W (Fig. 8b, left; large: $t(333) = 0.47$, $p = 0.643$; small: $t(333) = 2.49$, $p = 0.013$, two-sample *t*-test) and in both large- and small-reward trials in Monkey S (Fig. 8b, right; large: $t(304) = 9.57$, $p < 0.001$; small: $t(304) = 6.60$, $p < 0.001$, two-sample *t*-test). These results suggest that suppression of CdN-projecting LPFC neurons reduces the negative response at the time of the saccade. Together, these

neural analyses show that CNO administration affects task-related responses and weakens their power.

As described above, we identified changes in behavioral parameters and event-related LFP components after CNO administration. Next, we examined the correlations between these changes. For this analysis, we calculated the difference in LFP power within the extracted ROIs and in behavioral parameters (success rate, peak saccade velocity, and saccade latency) before and after CNO/VEH administration for each session. As a result, no correlation was strong enough to be significant after correction for multiple comparisons (Supplementary Fig. 12).

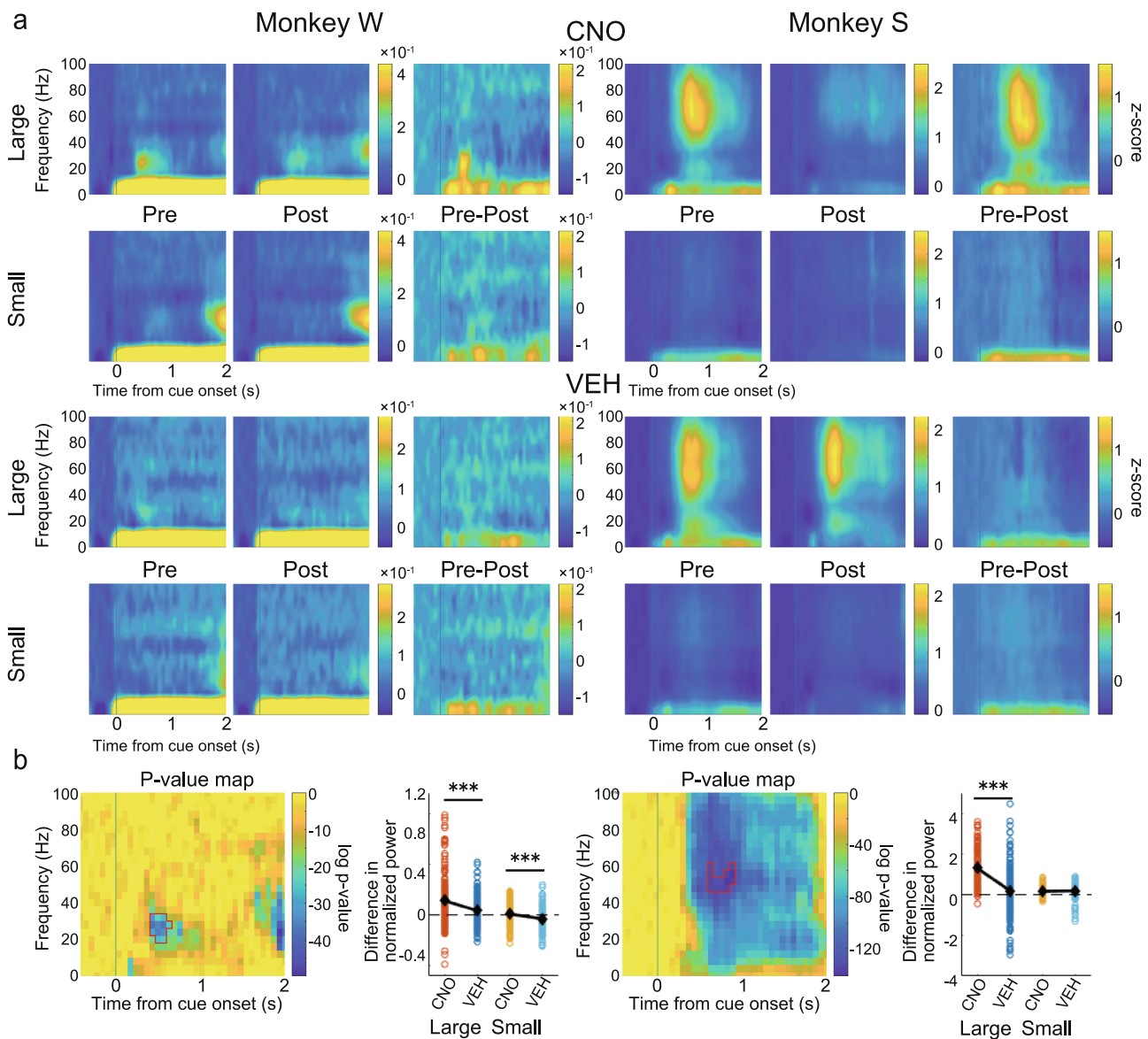

**Fig. 5 Effects on spectrogram power in the CdN after cue onset. a** Normalized and averaged spectrograms of LFPs from electrodes in the CdN for Monkey W (Left) and Monkey S (Right). All features in this panel are the same as in Fig. 4a. **b** For each monkey: left: *p*-value map to extract ROIs; right: comparison of effects of CNO/VEH administration (Pre–Post). All features in this panel are the same as in Fig. 4b. CdN caudate nucleus, CNO clozapine N-oxide, VEH vehicle, LFP local field potential, ROI region of interest.

## Discussion

In the present study, we used chemogenetic double virus transduction to selectively express an inhibitory DREADD (hM4D$_i$) in CdN-projecting LPFC neurons in macaque monkeys in order to elucidate the function of the LPFC-CdN pathway at a causal level. After extensive virus injections, a large number of mCherry-positive neurons indicating co-expression of hM4D$_i$ was identified in the LPFC. After CNO administration, consecutive errors occurred more frequently than after VEH administration, and the percentage of incomplete sessions in which the monkeys were unable to return to their normal performance significantly increased. With respect to eye movement behavior, a higher peak saccade velocity and shorter latency were observed after CNO administration selectively in saccades made to obtain a reward. Time-frequency analysis using LFPs showed that the positive response to the cue stimuli, which signaled the reward amount and target position, became smaller only after CNO administration in both the LPFC and CdN. The same analysis applied to the LFPs after the go signal revealed a reduced negative response during saccades only after CNO administration. These results confirm the successful induction of changes in behavioral and neural activity by the selective suppression of CdN-projecting LPFC neurons.

What do the observed behavioral changes tell us about the function of the LPFC-CdN pathway? While the LPFC has long been shown to be associated with working memory[62–64] and inhibitory control[65–68], the LPFC-CdN pathway has been noted to be important in certain subfunctions. Some previous studies have suggested that it is responsible for input and output information gating to working memory[20–23]. Input and output gating refers to the mechanism by which appropriate information is selected to be retained in the working memory or read from the working memory, respectively, to be used in a subsequent action. If the LPFC-CdN pathway is indeed involved in these mechanisms, then suppression of CdN-projecting LPFC neurons should result in working memory dysfunction. Thus, in the 1DR task, the information stored in working memory would be misrepresented

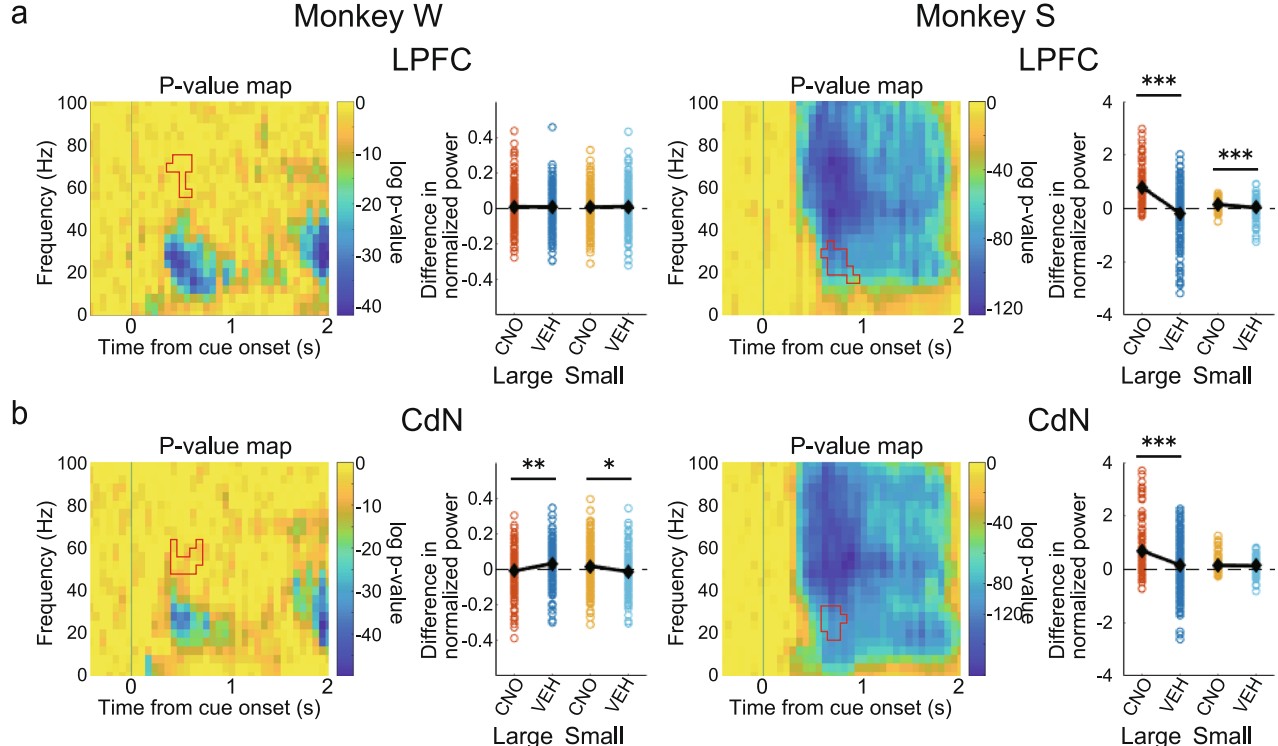

**Fig. 6 CNO effects after cue onset on different frequency bands.** Time-frequency analysis using ROIs exchanged between two monkeys to study the effect of CNO administration on different frequency bands. **a** CNO effects in the LPFC. For each monkey: left: *p*-value map and the ROI identified in the spectrograms of the other monkey with the time adjusted to the onset of the original ROI; right: comparison of effects of CNO/VEH administration (Pre–Post) using the other monkey's ROI depicted in the left panel. **b** CNO effects on the CdN. All features in this panel are the same as in Fig. 4b. CdN caudate nucleus, CNO clozapine N-oxide, VEH vehicle, LPFC lateral prefrontal cortex, ROI region of interest.

in behavior, resulting in increased saccades toward the opposite direction. However, no such increase in misdirected saccades was observed in the monkeys' behavior after CNO administration. Notably, in the 1DR task used in this study, only two target positions were used on the left and right sides of the horizontal direction, and the delay time was relatively short (~1000 ms); thus, the load on the working memory was not so heavy. It was reported, however, that inactivation of the macaque LPFC can induce working memory disruption with an approximately 1-s delay period[69].

Another group of studies has suggested that the LPFC-CdN pathway is associated with the inhibitory control of impulsive behavior[11–19]. Although inhibitory control has diverse meanings[70], the LPFC-CdN pathway has been reported to be linked to delay gratification in many of above studies. Delay gratification refers to the ability to withdraw immediate small rewards in order to obtain future large rewards and is deeply associated with patience, self-control, and willpower. Several diffusion tensor imaging (DTI) studies including human participants have shown that stronger connectivity between the LPFC and CdN is associated with higher delay gratification and a lower discount factor[11,15–17,19]. A similar result has also been obtained in a DTI study in chimpanzees[13]. This concept of inhibitory control in the sense of patience is also related to reward impulsivity. Low delay gratification was shown to be correlated with oversensitivity to immediate rewards or valuable stimuli[71]. If this hypothesis is correct, then suppression of CdN-projecting LPFC neurons should result in enhanced impatient behavior and impulsive behavior toward rewards. In the present study, the selective suppression of this pathway increased the percentage of incomplete sessions and frequency of consecutive errors. The increase in errors was mainly caused by fixation breaks around

the cue presentation, which indicates that the monkeys were impatient in the long-term performance of the task. In addition, we observed faster peak velocity and shorter latency in saccades after the go signal after CNO administration, suggesting an increase in impulsivity when the reward becomes close.

Previous studies have shown that the LPFC is related to more general functions, such as effort or motivation (e.g., ref. [72]). Suppression of the LPFC-CdN pathway may have resulted in behavioral changes by affecting the motivation of monkeys. Decay in motivation is associated with increased saccade latency[73]. Our results showed that saccade latency gradually decreased after CNO administration during both complete and incomplete sessions. Saccade latency rose sharply in a few dozen trials before the end, especially during the incomplete sessions, and the success rate began to decrease shortly before this rise. These observations suggest that the direct effect of CNO was the loss of inhibitory control manifested in the gradual decrease in saccade latency. In addition, the loss of inhibitory control caused an increase in errors, which may have led to a decrease in motivation, indicated by a subsequent rapid increase in saccade latency. Thus, our results indicate that the primary effect of suppression of the LPFC-CdN pathway is a dysfunction of inhibitory control, and the decrease in motivation is secondary. In summary, the observed behavioral changes caused by the suppression of CdN-projecting LPFC neurons are consistent with the hypothesis that the LPFC-CdN pathway is responsible for inhibitory control function in the sense of patience or self-control. Unlike previous MRI studies, we tested the inhibitory control function of the LPFC-CdN pathway through causal intervention in a pathway-selective manner.

Further, we recorded multichannel LFPs simultaneously from the LPFC and the ipsilateral CdN while the monkeys performed the 1DR

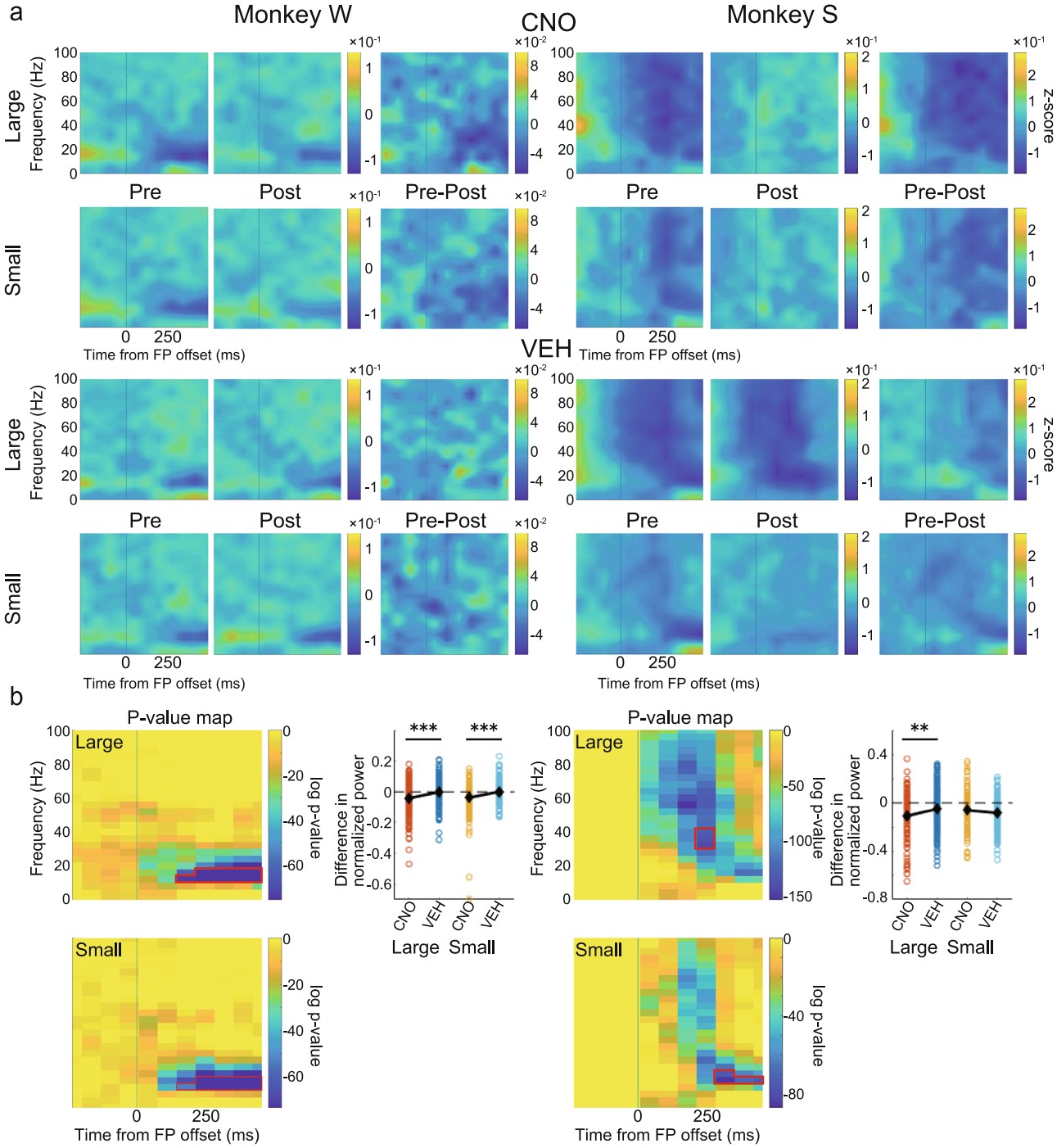

**Fig. 7 Effects on spectrogram power in the LPFC after the go signal. a** Normalized and averaged spectrograms of LFPs from electrodes in the LPFC for Monkey W (left) and Monkey S (right). Time 0 refers to the FP offset. All other features in this panel are the same as in Fig. 4a. **b** For each monkey: left: *p*-value maps made to extract ROIs with a negative component separately for large- and small-reward trials. Color represents log *p*-value obtained by comparing each time-frequency data point with the corresponding baseline data (−500 to 0 ms before cue onset); right: comparison of the effects of CNO/VEH administration. All features in this panel are the same as in Fig. 4b. CNO clozapine N-oxide, VEH vehicle, FP fixation point, LFP local field potential, LPFC lateral prefrontal cortex, ROI region of interest.

task. In terms of behavior, CNO administration caused increased errors, especially around the cue presentation and increased the impulsivity of saccades. Thus, in the LFPs recorded from the LPFC and CdN, we expected that the CNO-induced suppression would result in signal attenuation of cue-related and saccade-related activities. To confirm this prediction, we used time-frequency analysis to examine changes in neural activity induced by CNO administration. Spectrograms showed a phasic positive response after cue

presentation. Previous studies have shown that LPFC neurons show a differential response to cues that signal different reward amounts[74,75]. Consistently, the magnitude of the cue response in spectrograms varied depending on the reward amount informed by the cue. This cue response was significantly attenuated in both the LPFC and CdN after CNO administration compared to that after VEH administration. We also observed a negative response after the go signal. This negative response was also reduced in both the LPFC and CdN after

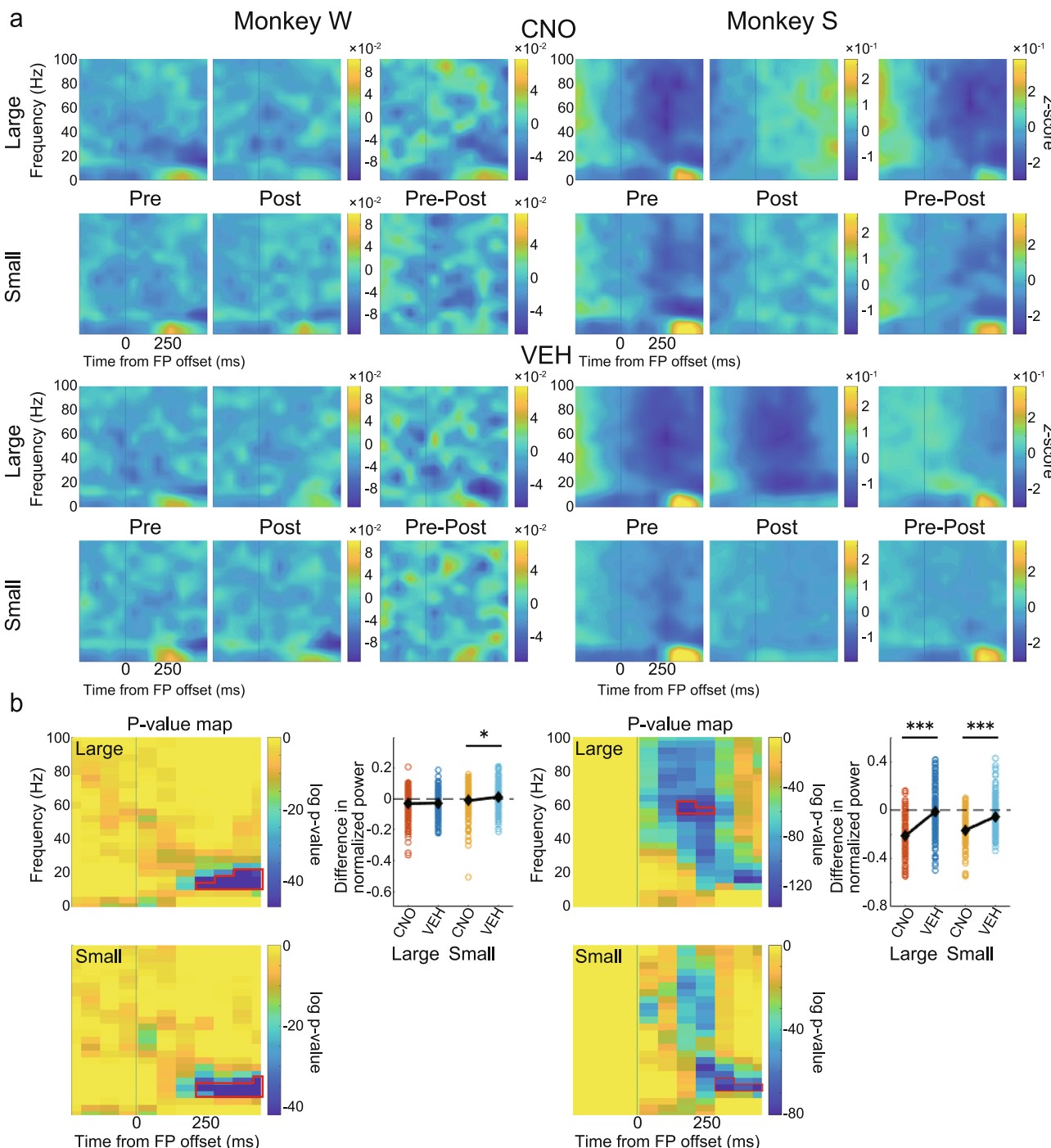

**Fig. 8 Effects on spectrogram power in the CdN after the go signal. a** Normalized and averaged spectrograms of LFPs from electrodes in the CdN for Monkey W (left) and Monkey S (right). All features in this panel are the same as in Fig. 7a. **b** For each monkey: left: *p*-value maps made to extract ROIs with a negative component; right: comparison of the effects of CNO/VEH administration. All features in this panel are the same as in Fig. 7b. CdN caudate nucleus, CNO clozapine N-oxide, VEH vehicle, LFP local field potential, ROI region of interest.

CNO administration compared to that after VEH administration. Although the LPFC is not directly involved in eye movement control, it may indirectly regulate eye movements through the formation of motor plans and online monitoring to facilitate their normal performance[76]. If so, it is possible that the negative signals in the LPFC observed during saccades are involved in the inhibitory control of eye movements, and that their attenuation leads to disinhibition and poor eye movement control. The increased impulsivity in saccades after CNO administration may be due to the attenuation of oculomotor regulation. These analyses indicate the successful

induction of changes in task-related neural activity following CNO administration. The reduced reward-dependent positive response after cue onset and the negative response in saccade timing may be responsible for the observed behavioral changes such as the increased consecutive errors and increased saccade impulsivity.

Time-frequency analysis revealed that the cue response was confined to the beta band in Monkey W but centered on the gamma band and extended to other frequency bands in Monkey S. After CNO administration, the cue response was consistently attenuated in the beta band in the two monkeys. In addition, it

was attenuated in the gamma band in Monkey S. By recording LFPs from the macaque PFC, previous studies have shown that signals in different frequency bands differentially contribute to information processing during task performance[77–79]. According to these studies, the gamma band is associated with the bottom-up processing of information, while the beta band is involved more with abstract top-down processing. The cue response in the gamma band observed only in Monkey S may be due to some individual difference in bottom-up information processing, although behavioral analysis did not differentiate this. In contrast, the cue response observed in the beta band, which was common between the two monkeys, may involve top-down information processing executed by the LPFC, which is considered to be at the top of the control hierarchy in the prefrontal network[80]. If this is the case, a decrease in this beta band cue response after CNO administration explains the loss of inhibitory control in the behavioral changes we observed.

In this study, doubly-transduced neurons expressing hM4D$_i$ were labeled with mCherry. By immunohistochemistry, we observed a large number of mCherry-positive neurons in the bilateral LPFC of the two monkeys. The retrograde vector incorporated eGFP as a reporter protein. Similar to a previous study[50], we observed a high frequency of eGFP-positive neurons in the bilateral CdN of Monkey S. However, we could not find eGFP-positive neurons in the CdN of Monkey W. Although the reason for this is unclear, multiple needle traces were visible in the bilateral CdN of Monkey W, indicating that the retrograde vector had been properly injected into the CdN. These observations show that we successfully expressed hM4D$_i$ selectively to neurons that compose the LPFC-CdN pathway.

Prefrontal circuits form a complex network that connects many brain regions. LPFC neurons innervate not only the striatum but also other cortical and subcortical areas[1,81]. Similarly, the striatum also receives projections not only from the PFC, but also from extensive cortical areas and the thalamus, hippocampus, amygdala, and midbrain dopamine areas[1,82]. However, conventional physical and pharmacological methods of controlling neural activity work indiscriminately against these diverse neural pathways. As a result, it becomes impossible to determine which pathway is involved in a specific function. Neural regulation by chemogenetics has similar caveats when performed through a single transduction and systemic administration of the ligand, and all current applications in macaque monkeys have been not pathway-selective[35–40]. In contrast, in this study, we combined chemogenetics via double virus transduction to selectively suppress LPFC neurons projecting to the CdN. This approach is not without its limitations. One of the problems is the control of neural transmission through the collateral branches. The chemogenetic double transduction method suppresses both the neural transmission from the CdN-projecting LPFC neurons to the CdN and that achieved by their collateral branches extending to other brain regions. For instance, among neurons that project from the cortex to the striatum, the pyramidal tract neurons extend their collateral branches beyond the striatum to the ipsilateral thalamus, subthalamic nucleus (STN), brainstem, and spinal cord[83,84]. Within these areas, the STN has been reported to be involved in some kind of inhibitory control[85,86] and is suggested to play a particularly important role in response inhibition, such as stopping behavior. If the suppression of CdN-projecting LPFC neurons simultaneously suppressed the input to the STN and decreased the ability of response inhibition, we would expect an increase in certain types of errors in the 1DR task such as saccadic fixation breaks induced by the cue stimulus immediately after the cue onset. However, this type of error was rare and its frequency did not increase compared with the frequency of other types of fixation breaks after CNO administration. To suppress

only the synaptic transmission of the target pathway and eliminate the effect on its collateral branches, a single transduction in the departure area (e.g., the LPFC) and microinjection of the ligand into the terminate area (e.g., the CdN) could be used. Another problem may arise in this case; this microinjection method limits the area where the ligand can be delivered compared to the wide ligand distribution achieved by systemic administration. In the case of macaques, which have a larger brain than rodents, this problem of a narrower area of drug delivery can be a particular drawback. A similar problem arises in optogenetics when pathway selectivity is achieved by photostimulation of local axon terminals. To overcome this kind of problem, the development of a method that can simultaneously deliver the ligand to a wide range of areas[87] or the development of a widespread photostimulation device with a sheet-like LED array[88] will be required.

We used systemic administration of CNO, the most widely used ligand, to control pathway-selected DREADDs. Recently, a rodent study[89] reported that CNO is converted to its reverse metabolite, clozapine, in vivo. Clozapine is then transported into the brain at a higher efficiency than CNO and acts on DREADDs as well as on off-target receptors (e.g., 5-HT, dopamine, or histamine receptors). Similar results have been reported in macaque monkeys[38,58]. We performed a behavioral test comparing the CNO and VEH conditions before the double virus transduction to assess the effect of CNO in the absence of DREADD expression. We found similar differences in the success rate and saccade accuracy for the CNO and VEH conditions as we did after double virus transduction. However, for other behavioral measures, including the proportion of incomplete sessions and consecutive errors, peak saccade velocity, and saccade latency, no effect of CNO alone was observed. Some of the behavioral changes we observed may be due to the effects of clozapine; however, it is unlikely that this is the case across the board. For a more complete prevention of off-target effects, it may be necessary to deliver CNO directly into the brain by microinjection or to use an effective and well-characterized ligand to replace CNO[38,90].

The LPFC-CdN pathway has been linked to a variety of psychiatric disorders such as schizophrenia, attention-deficit/hyperactivity disorder, obsessive-compulsive disorder, and drug addiction[70,91]. Abnormalities in inhibitory control, such as increased impulsivity and decreased delay gratification, have been frequently reported in these psychiatric disorders, which is consistent with the findings of this study. Compared to optogenetics, which requires the implantation of probes for photostimulation, chemogenetics works on artificial receptors only by the oral or subcutaneous administration of ligands, once DREADDs are expressed in the brain using viral vectors. In this respect, chemogenetics is less invasive and easier to implement in clinical applications[92]. In therapeutic applications, for example, excitatory DREADDs can be used to increase the activity of neurons in the LPFC-CdN pathway in order to ameliorate the hypoactivity of the LPFC seen in schizophrenia. The current study achieved the chemogenetic control of pathway-selective neuronal activity in the macaque brain, which is close to that of humans, and sets the basis for the development of new treatments for psychiatric disorders associated with functional abnormalities in specific neural pathways.

## Methods

**Subjects**. Two male Japanese monkeys (*Macaca fuscata*) were used as experimental subjects (Monkey W, 7 y, 9.0 kg, and Monkey S, 8 y, 7.5 kg). All experimental protocols in this study were approved by the Animal Care and Use Committee and the Safety Committee for Genetic Modification Research at Tamagawa University and were in accordance with the National Institutes of Health's Guide for the Care and Use of Laboratory Animals. The monkeys were purchased from the National Bioresource Project by MEXT, Japan. The monkeys

were kept in individual primate cages in an air-conditioned room where food was available *ad libitum*. Each cage was equipped with a platform along the middle of the wall, which enabled the monkeys to freely move up and down. A computer screen was placed in front of the cages to present the slide show of natural scenes to the monkeys for environmental enrichment. The body weight and appetite of the monkeys were checked, and vegetables and fruits were provided daily.

**Surgery**. The monkeys were anesthetized by intramuscular injections of ketamine hydrochloride (10 mg/kg) and xylazine (1 mg/kg) and maintained at a state of general anesthesia with isoflurane (1.0–2.0%). After the skull was exposed, a head holder was attached to the dental acrylic head implant, which was fastened to the skull by acrylic screws. Recording chambers (tilted 45° laterally in Monkey W and 30° in Monkey S in the coronal plane) were placed stereotaxically and secured with dental acrylic. A hole was drilled through the skull inside the recording chamber to allow cannula and electrode insertion. The center of each chamber was located near the posterior tip of the principal sulcus so that the injection cannulas and electrodes could reach both the LPFC and the head and body of the CdN. No behavioral abnormalities, loss of appetite, loss of weight, or other abnormalities were observed in either monkey during the experiment.

**Viral vector construction**. AAV5-hSyn-DIO-hM4D$_i$-mCherry (titer: $5.0 \times 10^{13}$ copies/mL) was purchased from the Gene Therapy Center Vector Core, University of North Carolina at Chapel Hill (http://www.med.unc.edu/genetherapy/vectorcore, no longer available, now, Addgene https://www.addgene.org/Bryan_Roth/). Briefly, hM4D$_i$-mCherry coding sequences were amplified by PCR, and the amplicons and an AAV vector carrying the human synapsin 1 promoter[93] were digested with NheI and AscI. The digestion products were ligated such that the coding regions of the fusion proteins were in a 3′ to 5′ orientation relative to the promoter. The final vectors were sequence verified and packaged in serotype 5.

A vector for neuron-specific retrograde gene transfer (NeuRet) was prepared by pseudotyping an HIV-1-based vector (titer: $1–3 \times 10^{13}$ copies/mL) as described previously[53–55] with some modifications. Briefly, the transfer plasmid (pCL20c-MSCV-nls-Cre-2A-eGFP) contained the cDNA encoding Cre recombinase fused to a nuclear localization sequence, 2A peptide, and enhanced green fluorescent protein (eGFP) downstream of the murine stem-cell virus promoter (MSCV). The envelope plasmid contained the cDNA encoding fusion glycoprotein type E (FuG-E) under the control of the cytomegalovirus enhancer/chicken β-actin promoter. HEK293T cells were transduced with the transfer, envelope, and packaging plasmids using the calcium-phosphate precipitation method. The cultured medium was harvested and filtered through a 0.45-μm Millex-HV filter unit (Millipore, Billerica, MA). The viral particles were centrifuged at $6000 \times g$ for 16–18 h and resuspended in phosphate-buffered saline (PBS). The particles were applied to a Sepharose Q FF ion-exchange column (GE Healthcare, Buckinghamshire, UK) and then concentrated by centrifugation through a Vivaspin Turbo filter (Vivascience, Lincoln, UK).

**Virus injection**. The mechanism by which the double virus transduction used in this study works has been described elsewhere[50]. Briefly, the technique utilizes a "local virus" that locally transduces the cell bodies of neurons and a "retrograde virus" that is retrogradely transported from neuronal axon terminals to the cell nuclei. The local viral vector incorporated the "Cre-On" FLEX double-floxed sequence, in which mCherry and hM4D$_i$ were included. The retrograde viral vector incorporated Cre-recombinase and eGFP. The local virus was injected into the bilateral LPFC, whereas the retrograde virus was injected into the bilateral CdN containing axon terminals of LPFC projection neurons. Using this method, hM4D$_i$ was selectively expressed in doubly-transduced LPFC neurons that project to the CdN.

During virus injection, the monkeys were seated in a primate chair with their heads fixed inside a sound-attenuated and electrically shielded room in a P2A-level experimental area. Before injection, we monitored the neural activities of the LPFC and CdN via extracellular recordings using a Multichannel Acquisition Processor (MAP) system (for Monkey W) or OmniPlex system (for Monkey S; Plexon, TX, USA) and identified the depth of gray matter for each recording track. In each recording session, the dura matter was first penetrated using a guide tube, and then, a tungsten electrode (0.5–2.6 MΩ; FHC, ME, USA) was lowered through a recording grid (holes: 0.6 mm diameter and 1.0 mm apart from center to center; Nakazawa, Chiba, Japan) using the NAN microdrive system (NAN instruments, Nazareth, Israel). Two different depths per track in the LPFC were chosen as injection sites for the local virus, typically 0.5 mm and 1.5 mm above the bottom of cortical neuron layers. Three different depths in the CdN were chosen as injection sites for the retrograde virus, typically 2.0 mm, 4.0 mm, and 6.0 mm below the first neuron we detected in the CdN. After identifying the gray matter and determining the injection sites for each track, the viruses were injected using a microsyringe (Ito, Shizuoka, Japan) connected to a 25-G (for the LPFC) or 31-G (for the CdN) needle. The needle was slowly lowered (0.2 mm/min) using the NAN microdrive system into the same track and stopped at 0.5 mm below the first injection site, which was the deepest among the target depths. After 1 min, the needle was withdrawn to the first injection site. The viral solution (2.0 μL) was then injected at a rate of 0.2 μL/min. The needle was maintained in place for an extra 15 min for diffusion and then withdrawn to the second injection site. The same procedure was repeated for all the remaining target depths. During the neural recording and waiting periods of virus injection, the monkeys performed the task described below to maintain their performance level.

**Behavioral procedure**. The monkeys were trained on the one-direction reward saccade (1DR) task, which is a memory-guided saccade task involving an asymmetric reward schedule. The task was performed inside a dark, sound-attenuated, and electrically shielded room. The monkeys were seated in a primate chair in front of a panel equipped with an array of green light-emitting diodes (LEDs; Monkey W) or a 22-inch LCD monitor (S2232W, EIZO, Ishikawa, Japan; Monkey S) with their heads fixed. The distance between the eyes and the panel or the display was 58 cm. A trial started with the appearance of a central fixation point (FP; green LED light with 0.4° diameter on the panel or white circle with 1.0° diameter on the monitor). A trial was counted as an abort error (fixation refusal) if the monkeys did not move their eyes to the FP for 5000 ms after its onset. After starting the fixation, the monkeys were required to fixate to the FP for 1000 ms. A cue (the same size and color as the FP) then appeared for 100 ms on either the left or right position with 10° eccentricity. The target position was determined pseudo-randomly using a Gellermann sequence. The disappearance of the FP after a variable delay period (1000 ± 250 ms) signaled the monkey to make a saccade to the previously cued position. The target reappeared 400 ms later and remained at the cued location for 150 ms. The saccade was judged to be correct if the eye position was within the "target window" (usually within 3° around the center of the target) when the target disappeared. If the monkey made a correct saccade, a reward of water and a success tone (1000 Hz) was delivered, and then the next trial started after an inter-trial interval (ITI: 4000 ± 2000 ms). Otherwise, no reward and an error tone (200 Hz) were delivered, followed by a time-out (2000 ms) in addition to the ITI. In the behavioral control tests performed before the double transduction, the variable delay period was set to 3000 ± 250 ms for both monkeys. After the long vector injection procedure, the monkeys became intolerable to the same delay period; thus, we decided to shorten it to 1000 ± 250 ms.

In each block of trials, one fixed direction was associated with a large reward (0.4 mL) and the opposite direction was associated with a small reward (0.1 mL). We applied a correction method as follows: if the monkeys made an error, the same trial was repeated. Therefore, the monkeys could not receive future large rewards without successfully performing less-motivated small-reward trials. The association between cue direction and reward size was pseudo-randomly changed in each block such that the large reward was associated with the left direction in two out of four successive blocks and with the right direction in the other two blocks. Each block consisted of 40 non-correction trials.

The task was controlled by the TEMPO system (reflective computing, MO, USA). Visual stimulus presentation for the monitor was programmed by a custom-made program using an application programming interface (OpenGL).

**Drug administration**. CNO at 5.0 mg/kg of body weight (BW; MedChemExpress, NJ, USA) was first dissolved in dimethyl sulfoxide (DMSO: 10 μL/mg) and then diluted with saline to a final volume of 0.8 mL/kg BW. A control solution (vehicle: VEH) was prepared with the same concentration of DMSO without CNO. At the start of a session, the monkeys were sat on a primate chair and anesthetized with isoflurane (5%) using a veterinary anesthesia mask. An intravenous line was then established from the leg. Thirty minutes after awakening, the monkeys first performed 160 trials (4 blocks) of the 1DR task. CNO or VEH was then administered intravenously. After CNO or VEH administration, the monkeys were required to complete an additional 720 trials (18 blocks).

**Recording and data acquisition**. Extracellular recordings of the local field potentials (LFPs) were conducted by linear-array multi-contact electrodes (U-probe, Plexon, TX, USA). Each electrode had 16 contacts (channels) with an inter-contact spacing of 100 or 150 μm. In each recording session, two U-probe electrodes were simultaneously inserted into the LPFC and the ipsilateral CdN. The dura matter was penetrated using stainless guide tubes, and U-probe electrodes were then advanced into the cortex through the guide tubes using the NAN microdrive system. A local reference was taken from the guide tubes close to the electrode contacts. The analog signals from each contact were split to extract spike and LFPs using Plexon MAP or OmniPlex systems. LFP signals were amplified (gain: 1000×), filtered with a passband of 0.1–200 Hz, and digitized at a sampling rate of 1 kHz. Eye movement was monitored by an infrared camera system at a sampling rate of 500 Hz (Monkey W: EyeLink2, SR research, Ontario, Canada; Monkey S: iView X Hi-Speed Primate, SMI, Teltow, Germany).

**Immunohistochemistry**. The protocol for immunostaining was basically the same as that used in a previous study[50]. After the end of the recording sessions, the monkeys were deeply anesthetized with an intravenous injection of sodium pentobarbital (70 mg/kg, i.v.) and transcardially perfused with 0.01 M PBS and then with 4% paraformaldehyde in 0.1 M phosphate buffer (pH 7.4). Brains were extracted and post-fixed in 4% paraformaldehyde overnight at 4 °C and cryoprotected through increasing gradients of sucrose (5, 10, and 20%). Frozen brains were then sliced into coronal sections at a thickness of 30 μm using a cryostat.

One in four successive sections was immunohistochemically stained. Free-floating sections were washed in PBS and permeabilized in PBS containing 0.3%

Triton X-100 (PBST). After blocking for 1 h in 3% normal goat serum in PBST containing 1% bovine serum albumin (BSA-PBST), sections were incubated with primary antibodies in BSA-PBST for 2 nights at 4 °C. eGFP was detected using a mouse anti-GFP antibody (1:250; Millipore, MA, USA), and mCherry was detected using a rabbit anti-RFP antibody (1:500; Abcam, Cambridge, UK). After washing in PBST, sections were incubated with secondary antibodies in BSA-PBST at room temperature for 4 h. eGFP was visualized with Alexa-488 labeled goat anti-mouse IgG (1:500; Molecular Probes, OR, USA), and mCherry was visualized with Alexa-568 labeled goat anti-rabbit IgG (1:500; Molecular Probes, OR, USA). After washing in PBS, sections were mounted on glass slides with Fluoromount (Diagnostic BioSystems, CA, USA).

mCherry and eGFP fluorescence was captured using a camera lucida attached to an epifluorescence microscope (BX51, Olympus, Tokyo, Japan) equipped with 10×, 20× and 40× objective lenses. The number of mCherry-positive cells in LPFC-containing slides was counted manually using an epifluorescence microscope. Images of Nissl-stained sections were obtained using a stereo microscope (EZ4E, Leica, Germany).

**Data analysis**. Off-line analysis was carried out using custom-made MATLAB programs (MathWorks, MA, USA). To assess the behavioral effect of the suppression of LPFC neurons projecting to the CdN, we divided the sessions into complete and incomplete according to the following criterion: a session was counted as incomplete if the monkeys performed 10 consecutive errors during the 720 trials after CNO/VEH administration (10 consecutive errors × 1 time) or more than 5 consecutive errors three times during the same period (more than 5 consecutive errors × 3 times). In such a session, the monkeys typically could not return to their normal performance. The correct rate was calculated considering the first 160 trials (before CNO/VEH administration: Pre) and the last 160 trials (after CNO/VEH administration: Post), including repeated error trials and correction success trials, in order to account for the effect on successive errors. Since the monkeys performed 160 trials before CNO/VEH administration, the number of trials included in Pre was set to 160, and the number of trials included in Post was set to the same number as in Pre. The first and second non-correction trials and (if any) subordinate correction trials after block change were discarded.

To calculate the frequency of errors, we first divided errors into four types, as follows: (1) fix refusal: abort error before fixation, (2) Pre cue: fixation break before the cue onset, (3) delay: fixation break during the delay period, and (4) saccade: saccade error after the go signal. The number of times each error was repeated was separately counted in each condition (Pre/Post in the CNO condition and Pre/Post in the VEH condition). Then, each value was divided by the relevant number of trials and further divided by the time duration associated with each error (fix refusal: 5 s, Pre cue: 1 s, delay: 1 s, saccade: 0.55 s). The frequency of errors indicated the number of errors per second.

We judged that an eye movement occurred if the velocity continuously exceeded the following threshold: maintained more than 30°/s for at least 10 ms. Eye movement was defined as a saccade if its peak velocity exceeded 200°/s. To calculate the peak saccade velocity, saccade latency, and saccade endpoint after the go signal, we included eye position data only from non-correction success trials. We excluded saccade data if the peak velocity was above the mean value plus two standard deviations (typically caused by the loss of eye tracking due to eye blink). Saccade latency was calculated as the time between the offset of the central FP and the beginning of a saccade eye movement (when its velocity exceeded 30°/s). The saccade endpoint was the mean eye position during 10–20 ms after the end of the last saccade eye movement (when its velocity fell below 30°/s). If more than two saccades were included during the go period (0–550 ms after the offset of the FP), we considered the first and last saccades for saccade latency and saccade endpoint, respectively. To calculate the peak saccade velocity and saccade endpoint at the initial fixation, we chose trials in which the eye position was outside the fixation window at FP onset.

Raw LFP data were first aligned to the cue onset. To remove channels with excessively low signal levels or high noise levels, we calculated the absolute mean of the raw data for each channel and discarded channels outside the following criteria: less than 10 μV or more than 100 μV. Before proceeding with the analysis, the line noise (50 Hz) was removed from the LFP data using the rmline function in Chronux (http://chronux.org/). To remove common noise (e.g., licking behavior or click noise) across channels simultaneously recorded from the same U-probe electrode, we performed principal component analysis for each trial using the pca function in MATLAB), discarded the first and the second principal components, and reconstructed LFP data using the remaining principal components.

**Spectrogram analysis**. To examine whether the power of LFPs recorded from the LPFC and CdN was affected by CNO/VEH administration, we first conducted a time-frequency analysis of the LFP power around cue onset. Spectrograms were calculated using the fast Fourier transformation (spectrogram function in MATLAB) to estimate the power of LFPs in the cue and delay periods. We used LFPs from the first and the last 160 non-correction trials of each session (Pre and Post) in this analysis. After normalizing spectrograms for each frequency using the mean power before the cue onset (from −500 to 0 ms) as the baseline, we selected channels showing significant cue-responses using a sliding window (10-Hz moving frequency bin, 10-Hz step, and 500-ms time bin fixed to 400–900 ms after the cue onset to cover strong cue responses; compared with corresponding baseline data,

paired samples *t*-test, Bonferroni-corrected, α = 0.05). To extract an appropriate time window as the time-frequency region of interest (ROI) in which the effect of CNO/VEH administration should be evaluated, we applied a cluster-based non-parametric test[59,60] to spectrograms obtained during the Pre period. We first divided spectrograms according to large- and small-reward conditions and then prepared a *p*-value map by comparing them (i.e., large- vs. small-reward conditions) with paired samples *t*-test for each time-frequency data point. This map was binarized such that time-frequency data points with *p*-values in the bottom 5% during the delay periods were set to 1, whereas the other data points were set to 0. Adjacent data points with the value 1 then clustered together, and the largest time-frequency cluster was defined as the ROI. Time points within the ROI were averaged and then statistically compared between Pre and Post separately for the CNO and VEH conditions. We next applied the same analysis to the LFP data aligned to FP offset (the go signal). The mean power before the cue onset (from −500 to 0 ms) was used as the baseline. To extract ROIs, we again applied a cluster-based nonparametric test during the Pre period. We generated a *p*-value map by comparing data points between baseline and each time frequency using one-sided paired samples *t*-test separately for large- and small-reward trials in order to extract characteristic negative components.

**Statistics and Reproducibility**. No statistical methods were used to predetermine sample sizes, but our sample sizes are similar to or larger than previous chemogenetic macaque studies[35,37,38,40]. Statistical analyses were conducted in Matlab (2017b). To test the proportion of incomplete sessions, saccades in the opposite directions after the cue offset, and fixation breaks produced by making a saccade to the cue direction after the cue onset, Fisher's exact test were performed. Wilcoxon rank-sum test was used to compare the frequency of consecutive errors between CNO and VEH conditions. To compare other behavioral parameters between Pre and Post conditions, we performed two-tailed, paired samples *t*-tests, and between CNO and VEH conditions, we performed two-tailed, unpaired Student's *t*-tests. We also conducted a two-way ANOVA using the main factors for the drug (CNO, VEH) and time (Pre, Post) associated with these behavioral parameters. To compare Pre/Post differences in LFP signals in time-frequency ROIs between CNO and VEH conditions, we also used two-tailed, unpaired Student's *t*-tests. Significance was set at α = 0.05 for all statistical tests. To examine the relationship between the Pre and Post changes in behavioral parameters and similar changes in LFP power, we calculated the Pearson correlations with Bonferroni correction.

**Reporting summary**. Further information on experimental design is available in the Nature Research Reporting Summary linked to this paper.

## Data availability

Essential source data for main figures and supplementary figures are included in Supplementary Data 1 and 2, respectively. Additional data supporting the findings of this study are available from the corresponding author upon reasonable request.

## Code availability

Matlab codes used in the analysis of this study are available from the corresponding author upon reasonable request.

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

## Acknowledgements
The authors are grateful to M. Koizumi for technical assistance with the experiments. This paper was proofread in English by a native speaker from Editage (https://www.editage.jp). This work was supported by Grant-in-Aid for Scientific Research on Innovative Areas (4303 and 4805), JSPS KAKENHI Grant Number JP18H03662 from MEXT of Japan, the National Natural Science Foundation of China (No: 11972159).

## Author contributions
M.O., T.K. and M.S. designed the animal experiments. M.O. and S.T. performed the animal surgeries. S.K. and K.K. developed and produced viral vector (NeuRet). M.O. performed the animal experiments. K.M-I. performed immunohistochemistry. M.O. and X.P. analyzed the data. M.O., T.K., and S.K. wrote the manuscript. All authors commented on the manuscript.

## Competing interests
The authors declare no competing interests.
