## [Peer Review File · Communications Biology]

Reviewers' comments:

Reviewer #1 (Remarks to the Author):

The manuscript by Oguchi et al describes experiments aimed at validating the function of the LPFC-CDN pathway at a causal level using pathway-specific chemogenetic manipulations in two macaques. The authors claim that they were able to successfully suppress neural activity in a pathway-specific manner, inducing both physiological and behavioral changes relevant to inhibitory control mechanisms. This is a well-written and well-designed study with important technical achievement and scientific findings.

Line 53: suggest to use another term instead of "inappropriate" for describing behavior. This is too subjective of a term.

Figure 1: can the authors show the images in panels c and d and e and f superimposed to each other? Also the legend contains an abbreviation for CNO but this is not referred to in the figure.

Suppl figures: the images in these panels are of too low resolution.

Figure 2 and elsewhere: I would suggest to use the term "transduction" instead of "infection"

Line 398: "should result in working memory dysfunction"

Line 518: This statement needs to be revised because the CSF does not constitute the brain parenchyma where DREADDs are found. CNO is biased towards CSF but not brain. Clozapine preferentially enters the brain and is not enriched in CSF.

Reviewer #2 (Remarks to the Author):

The development of chemogenetic approaches such as DREADDs in non-human primates has been a major challenge in neuroscience. Here Oguchi, Sakagami, and colleagues report on the development and use of pathway specific DREADDs to alter behavior and physiology in the projections from the dorsolateral prefrontal cortex (dlPFC) to the caudate in macaques. They show that inhibiting activity in this pathway using the hM4Di DREADD impairs behavior in subtly different ways in two macaques performing the now classic delayed-response saccade task. Further, the change in behavior was related to specific changes in local field potential activity in both dlPFC and caudate. Histological analysis of the brain tissue confirmed that the injections of virus has indeed labeled specific projection neurons.

Overall, manuscript is clearly written, the findings are potentially important to the field, and this challenging work is well executed. However, I have several critical concerns on the study design and the interpretation of the data, which need to be clarified in the text.

Major comments:

In this study, the authors used CNO (5mg/kg) as an actuator for hM4Di. This is a reasonable approach but needs careful controls because CNO metabolizes to clozapine and this can have off target effects in the brain. The authors are fully aware of the issue and actually have behavioral data from animals before DREADD infection (Fig. 2), yet the data were not fully reported. Specifically, changes in saccade parameters by CNO administration before injection of virus need to be reported, as it is the critical data to determine if dlPFC-caudate interaction is involved in impulsive control. These control analyses are essential because clozapine on its own could increase impulsivity through dopaminergic mechanisms.

Related to the first comment, the authors need more careful discussion on the effects of CNO itself in the data. In the text, the authors claimed "it is unlikely that our results are due to an off-target effect of clozapine" but it contradicts to the data shown in Figure 2c,d in which CNO induced the reduction of success rate preferentially in small reward trials in both monkeys even before

infection. In this regard, the effect of CNO itself should partially explain the results.

Further, it is also possible that the dIPFC-caudate is actually involved in more general function, such as motivation or effort. The result in Figure 2b may fit this theory, because monkeys could have reduced performance in the CNO condition due to a faster decay in motivation. Indeed, a number of studies have reported that a role for dIPFC in motivation or effort (e.g., Soutschek and Tobler (2020), "Causal role of lateral prefrontal cortex in mental effort and fatigue"). So, that being said, I'm interested to know if there any changes in performance, such as reaction time over the course of the sessions that might be indicative of a change in motivation.

The analyses of the behavioral data are mostly appropriate, but the authors are adding more statistical tests than are required. For instance, they often conduct separate t-test to confirm effects of reward and CNO, when a single 2 by 2 analysis would suffice (e.g. line 166-173. Conducting these analyses would also mean that the authors could report interaction effects between CNO and reward instead of having to infer these.

The authors conduct a control analysis on effect of contralateral and ipsilateral presentation of stimuli on activity in dIPFC (Lines 319-326). This is useful but I wonder if they should expand their control analyses to the presentation of the fixation spot in the task and the response of oscillations to the presentation of the fixation spot on and off drug. This type of analysis is useful as it would enable the authors to show/prove that the effects of CNO are specific to stimuli that are essential for working memory as opposed to a general effect on stimulus induced neural activity in dIPFC/caudate.

I am interested to know if the changes in LFP power seen in the two monkeys are correlated with the corresponding change in behavior in both animals. If LFP changes induced by CNO reflected the changes in behaviors, modulation level of LFP signal should correlate to the reduction of correct performance or other changes in saccadic parameters. Even if there is not a relationship, I think it is worth reporting this analysis in the revised manuscript.

The neurophysiology analysis focusses exclusively on LFPs in single areas. This misses two wider points. LFPs are, in certain frequency bands, thought to reflect both local processing AND inputs to an area. In that regard it would be interesting to know how multi-unit activity in either dIPFC or caudate are altered by the administration of CNO. These data should be available for analysis and would be an elegant demonstration that DREADDs alter spiking activity. Second, I was surprised that the authors did not look at metrics of inter-area communication between dIPFC and caudate. Measures of synchrony/coherence feel appropriate here. Neither of these analyses are essential to the manuscript but would certainly improve it.

Minor comments:

The physical size (ml) of the large and small rewards should be reported in the main text and figure legends. At the moment this information is hard to find.

Figures are very confusing due to the choice of color. Specifically, reddish colors are used for CNO condition and blueish colors are used for VEH condition in main figures but the similar colors are used to highlight different contrasts in the other figures (for examples, Pre vs Post in Figure 3b-d, and Complete vs Incomplete in Figure 2b). Consider revising these and making them consistent across the manuscript.

Figure 2d is very confusing. At a glance it indicates that CNO increases error rates in before and after viral infusions. However, the data the authors emphasize here is the difference in average number of consecutive errors. Why not make this in a density plot and mark the average number of consecutive errors on it for each condition?

Rebuttal Letter

Response to reviewer #1

Line 53: suggest to use another term instead of “inappropriate” for describing behavior. This is too subjective of a term.

The word “inappropriate” has been changed to “impulsive,” which has a less subjective meaning, at least, in the context of decision-making research (Lines 37, 53, 91, and 456).

Figure 1: can the authors show the images in panels c and d and e and f superimposed to each other? Also the legend contains an abbreviation for CNO but this is not referred to in the figure.

The images created by overlapping panels c and d and e and f have been included in Fig. 1, and relevant texts in the legend have been revised and added. The abbreviation for CNO in the legend has been removed.

Suppl figures: the images in these panels are of too low resolution.

We have attached a high-resolution PDF file of the supplementary figures. Please look at this.

Figure 2 and elsewhere: I would suggest to use the term “transduction” instead of “infection”

The words “infection” and “infect” in the manuscript have been changed to “transduction” and “transduce.”

Line 398: “should result in working memory dysfunction”

We have fixed it as instructed.

Line 518: This statement needs to be revised because the CSF does not constitute the brain parenchyma where DREADDs are found. CNO is biased towards CSF but not brain. Clozapine preferentially enters the brain and is not enriched in CSF.

Taking into account a related comment from Reviewer #2, the reference to CNO and clozapine concentrations in the CSF has been removed, and the text in this part of the manuscript has been

substantially revised (Line 574 onwards). We have acknowledged that CNO may have affected some of the behavioral changes (success rate and saccade accuracy; see Fig. 2 and Supplementary fig. 5), but we have emphasized that other behavioral changes could not be explained by the effects of CNO (or clozapine).

Response to reviewer #2

Major comments:

In this study, the authors used CNO (5mg/kg) as an actuator for hM4Di. This is a reasonable approach but needs careful controls because CNO metabolizes to clozapine and this can have off target effects in the brain. The authors are fully aware of the issue and actually have behavioral data from animals before DREADD infection (Fig. 2), yet the data were not fully reported. Specifically, changes in saccade parameters by CNO administration before injection of virus need to be reported, as it is the critical data to determine if dlPFC-caudate interaction is involved in impulsive control. These control analyses are essential because clozapine on its own could increase impulsivity through dopaminergic mechanisms.

The results of the eye movement analysis before the double virus transduction have been included in Supplementary fig. 5, and the relevant texts were added to the relevant parts (Line 248 and Line 284 onwards). We found that saccade accuracy worsened with the CNO condition compared with the VEH condition, even before transduction. It is difficult to say that the effect on saccade accuracy after transduction in Monkey W was due to the suppression of the LPFC-CdN pathway; therefore, we have deleted the result of saccade accuracy analysis at the FP onset (in the section “Effects of CNO administration on eye movement behavior at different time periods” and Supplementary figure 6) and the description of the effect on saccade accuracy from the Discussion session.

Related to the first comment, the authors need more careful discussion on the effects of CNO itself in the data. In the text, the authors claimed “it is unlikely that our results are due to an off-target effect of clozapine” but it contradicts to the data shown in Figure 2c,d in which CNO induced the reduction of success rate preferentially in small reward trials in both monkeys even before infection. In this regard, the effect of CNO itself should partially explain the results.

Taking into account a related comment from Reviewer #1, we have made a substantial modification of the description of the clozapine effect in the Discussion (Line 574 onwards). We

have acknowledged that CNO may have affected some of the behavioral changes (success rate and saccade accuracy; see Fig. 2 and Supplementary fig. 5) but emphasized that other behavioral changes could not be explained by the effects of CNO (or clozapine).

Further, it is also possible that the dlPFC-caudate is actually involved in more general function, such as motivation or effort. The result in Figure 2b may fit this theory, because monkeys could have reduced performance in the CNO condition due to a faster decay in motivation. Indeed, a number of studies have reported that a role for dlPFC in motivation or effort (e.g., Soutschek and Tobler (2020), “Causal role of lateral prefrontal cortex in mental effort and fatigue”). So, that being said, I’m interested to know if there any changes in performance, such as reaction time over the course of the sessions that might be indicative of a change in motivation.

We analyzed more detailed temporal changes in the saccade latency and success rate separately for the complete and incomplete sessions (Line 244 onwards and Supplementary fig. 4). The results showed that the saccade latency gradually decreased over several hundred trials after CNO administration and sharply increased near the end of the incomplete sessions. We also found that this sharp increase was preceded by a decrease in the success rate during incomplete sessions. We have added an interpretation of these results to the Discussion (Line 473 onwards), where we have insisted that the effect of the suppression of the LPFC-CdN pathway is the loss of inhibitory control, and the effect on motivation is, at, most, secondary.

The analyses of the behavioral data are mostly appropriate, but the authors are adding more statistical tests than are required. For instance, they often conduct separate t-test to confirm effects of reward and CNO, when a single 2 by 2 analysis would suffice (e.g. line 166-173). Conducting these analyses would also mean that the authors could report interaction effects between CNO and reward instead of having to infer these.

We have added the results of the ANOVA analyses of the effect of the interaction between the drug (CNO, VEH) and time (Pre, Post) in the parts reporting the success rate and saccade peak velocity, latency, and accuracy. Note that the interaction effect is not necessarily significant even when the T-test shows there is a difference between the Pre and Post effects with the CNO condition but not the VEH condition.

The authors conduct a control analysis on effect of contralateral and ipsilateral presentation of stimuli on activity in dlPFC (Lines 319-326). This is useful but I wonder if they should expand their control analyses to the presentation of the fixation spot in the task and the response of

oscillations to the presentation of the fixation spot on and off drug. This type of analysis is useful as it would enable the authors to show/prove that the effects of CNO are specific to stimuli that are essential for working memory as opposed to a general effect on stimulus induced neural activity in dlPFC/caudate.

We analyzed the LFP data aligned with the onset of the fixation point. As a result, there was no clear response in either monkey as there was after the cue onset (we added these spectrograms as Supplementary fig. 11 and relevant texts at Line 368 onwards). We applied a cluster analysis to these spectrograms but did not find any meaningful ROIs between the two monkeys during the same period. Our experimental results showed that CNO administration to the doubly transfected monkeys affected the inhibitory control but not the working memory in the first place. Therefore, even if the neural responses to the FP onset, which were not related to memory, were altered by CNO administration, this would not affect our results.

I am interested to know if the changes in LFP power seen in the two monkeys are correlated with the corresponding change in behavior in both animals. If LFP changes induced by CNO reflected the changes in behaviors, modulation level of LFP signal should correlate to the reduction of correct performance or other changes in saccadic parameters. Even if there is not a relationship, I think it is worth reporting this analysis in the revised manuscript.

We calculated the correlations between the changes in LFP power within the extracted ROIs (averaged over 16 channels per recording site) and changes in behavioral parameters (success rate, peak saccade velocity, and saccade latency) before and after CNO/VEH administration. The results showed that none of these correlations was significant after correction for multiple comparison. The results are presented in Supplementary fig. 12, and the corresponding text has been added after Line 417 onwards. These results were almost the same even when the CNO and VEH conditions were calculated separately.

The neurophysiology analysis focusses exclusively on LFPs in single areas. This misses two wider points. LFPs are, in certain frequency bands, thought to reflect both local processing AND inputs to an area. In that regard it would be interesting to know how multi-unit activity in either dlPFC or caudate are altered by the administration of CNO. These data should be available for analysis and would be an elegant demonstration that DREADDs alter spiking activity.

We admit that it would be interesting to analyze how CNO administration affected single- and multi-unit activities recorded from the LPFC and the CdN. We plan to perform such analyses in

the future, but it would take a long time to perform them carefully. Instead, we focused on LFP signals in the high-frequency band. Previous studies have shown that signals in the high-frequency band of LFP reflect multi-unit activity (Ray et al., 2008, <https://doi.org/10.1523/JNEUROSCI.1588-08.2008>; Ray and Maunsell, 2011, <https://doi.org/10.1371/journal.pbio.1000610>; Buzsaki et al., 2012, <https://doi.org/10.1038/nrn3241>). Therefore, we analyzed the LFPs in the high gamma band (100-300 Hz) as a surrogate for multi-unit activity. We found that the high gamma signal showed a characteristic cue response immediately after cue onset (Response fig. 1a attached to this letter). For the case of Monkey S, a large effect of the cue response observed under low gamma (Figs 4 and 5) was observed in the later period, which was distinct from this faster signal. To examine the activity in response to cue presentation, we averaged the data from 0 to 200 ms after the cue onset at 100-300 Hz and compared the Pre/Post differences between the CNO and VEH conditions. The results showed that the Pre/Post difference was significantly larger for the CNO than for the VEH conditions only in the small reward trials from the LPFC in Monkey W (Response fig. 1b). No significant differences were found for any of the other conditions. The results for two monkeys were not consistent and not significant enough, and they were not included in the revised manuscript.

Second, I was surprised that the authors did not look at metrics of inter-area communication between dlPFC and caudate. Measures of synchrony/coherence feel appropriate here. Neither of these analyses are essential to the manuscript but would certainly improve it.

To analyze the change in functional connectivity between the LPFC and CdN, we calculated the coherence (a measure of the trial-to-trial consistency of the phase difference between two signals at a given time) between the pairs of LPFC and CdN channels and created time-frequency coherograms (Response figs. 2a and 3a). We used these coherograms to conduct the same cluster analysis as the spectrograms, and the Pre/Post differences were compared between the CNO and VEH conditions. The results showed that there were significant differences between the CNO and VEH conditions in Monkey W after both the cue onset and the go signal in the small reward trials, but there was no significant difference in any conditions in Monkey S (Response figs. 2b and 3b). The periods for the ROIs extracted after the FP offset for the two monkeys were not consistent. The statistical results were also not consistent, and they were not included in the revised manuscript.

Minor comments:

The physical size (ml) of the large and small rewards should be reported in the main text and

figure legends. At the moment this information is hard to find.

We have inserted information about the amount of rewards in Fig. 2a and the corresponding legend.

Figures are very confusing due to the choice of color. Specifically, reddish colors are used for CNO condition and blueish colors are used for VEH condition in main figures but the similar colors are used to highlight different contrasts in the other figures (for examples, Pre vs Post in Figure 3b-d, and Complete vs Incomplete in Figure 2b). Consider revising these and making them consistent across the manuscript.

In Fig. 2b, we changed complete and incomplete to yellowish and grayish. In Fig. 3b-d, and Supplementary fig.3a and 6b-c, we changed the color scheme of the CNO and VEH for consistency with the other figures.

Figure 2d is very confusing. At a glance it indicates that CNO increases error rates in before and after viral infusions. However, the data the authors emphasize here is the difference in average number of consecutive errors. Why not make this in a density plot and mark the average number of consecutive errors on it for each condition?

As reviewer #2 pointed out, the original figure was difficult to understand because the values used in the statistical tests were not explicit. To make it easier to understand, the figure and the corresponding legend were changed to compare the average length of the consecutive errors for each condition (see Fig. 2d).

Response fig. 1 | The effect on high gamma band signal (100-300 Hz)

Response fig. 2 | Effects on coherogram after cue onset

Response fig. 3 | Effects on coherogram after go signal

REVIEWERS' COMMENTS:

Reviewer #1 (Remarks to the Author):

The authors have done a good job of addressing my comments. Thank you.

Reviewer #3 (Remarks to the Author):

The authors have done a good job responding to my concerns. The additional analyses that they presented and alterations to the manuscript have strengthened this paper. No further comments.